**Patterns of longer-term climate change effects on CO$_2$ efflux from biocrusted soils differ**
**from those observed in the short-term**
Anthony Darrouzet-Nardi[1], Sasha C. Reed[2], Edmund E. Grote[2], Jayne Belnap[2]
[1]University of Texas at El Paso, 500 W. University Ave., El Paso TX 79912 USA
[2]U.S. Geological Survey, Southwest Biological Science Center, Moab, UT 84532 USA
Correspondence: Anthony Darrouzet-Nardi (ajdarrouzetnardi@utep.edu)
**Abstract.** Biological soil crusts (biocrusts) are predicted to be sensitive to the increased
temperature and altered precipitation associated with climate change. We assessed the effects of
these factors on soil carbon dioxide ($CO_2$) balance in biocrusted soils using a sequence of
manipulations over a nine-year period. We warmed biocrusted soils by 2 and, later, by 4 °C to
better capture updated forecasts of future temperature at a site on the Colorado Plateau, USA.
We also watered soils to alter monsoon-season precipitation amount and frequency, and had
plots that received both warming and altered precipitation treatments. Within treatment plots, we
used 20 automated flux chambers to monitor net soil exchange (NSE) of $CO_2$ hourly, first in
2006-2007 and then again in 2013-2014, for a total of 39 months. Net $CO_2$ efflux from
biocrusted soils in the warming treatment increased a year after the experiment began (2006-
2007). However, after 9 years and even greater warming (4 °C), results were more mixed, with a
reversal of the increase in 2013 (i.e., controls showed higher net $CO_2$ efflux than treatment plots)
and with similarly high rates in all treatments during 2014, a wet year. Over the longer-term, we
saw evidence of reduced photosynthetic capacity of the biocrusts in response to both the
temperature and altered precipitation treatments. Patterns in biocrusted soil $CO_2$ exchange under
experimentally altered climate suggest that (1) warming stimulation of $CO_2$ efflux was
diminished later in the experiment, even in the face of greater warming and (2) treatment effects
on $CO_2$ flux patterns were likely driven by changes in biocrust species composition and by
changes in root respiration due to vascular plant responses.

## 1 Introduction

Soils with active biological soil crust (biocrust) communities are essential components of dryland ecosystems worldwide and are also one of the most sensitive components of drylands to climate change (Ferrenberg et al., 2017; Reed et al., 2016). Given the vast and growing global extent of dryland regions (Safriel et al., 2005; Prăvălie, 2016), the response of biocrusts to major global change phenomena, such as climate change, may be an important aspect of the overall response of Earth's ecosystems. In particular, due to the potential for dryland feedbacks to future climate (Poulter et al., 2014; Ahlström et al., 2015; Rutherford et al., 2017), a key parameter to consider as dryland ecosystems warm is carbon (C) balance, specifically carbon exchange of biocrusted soils. Dryland soils are characterized by low soil organic matter that is negatively correlated with aridity across many drylands (Delgado-Baquerizo et al., 2013) and there is an association between C loss and the phenomenon of desertification (Lal, 2004). Drylands can also show large year-to-year variation in C fluxes that are relevant for explaining global-scale fluxes (Ahlström et al., 2015; Poulter et al., 2014; Biederman et al., 2017). At the ecosystem scale, biocrusted soils within drylands are often substantial contributors to both C uptake (Elbert et al., 2012) and ecosystem respiration (Castillo-Monroy et al., 2011). At the organism scale, the viability of biocrust is linked to their ability to maintain a positive C balance among hydration-desiccation cycles (Grote et al., 2010; Coe et al., 2012; Oliver et al., 2005). Despite the importance of C balance to understanding biocrust function and dryland ecosystem feedbacks to global change, few studies have addressed how biocrust soil $CO_2$ fluxes will respond to changing temperature and precipitation.

Carbon balance in biocrusted soils includes not only the activities of the biocrusts themselves, but also the activities of subsurface vascular plant roots and soil heterotrophic microbes. Considering biocrusted soils together with the function of adjacent vascular plants is important given that there is increasing evidence for biotic connections, possibly mediated by fungi, between these functional groups (Green et al., 2008) and for linkages in plant-soil C cycle responses to warming. For example, at another site on the Colorado Plateau, measurements of plant photosynthesis, coupled with spot measurements of soil respiration under plant canopies, showed plant photosynthetic rates were tightly coupled to soil respiration rates, with both showing reduced fluxes in response to warming during the spring when plants are most active (Wertin et al., 2017). While these patterns could be the result of independent climate controls,

such as temperature and moisture, on each individual flux, vascular plant C allocation to roots
and heterotrophs belowground or biotic connections between biocrust organisms and vascular
plants could also help explain the coupling between above- and belowground $CO_2$ fluxes.

In addition to affecting soil C balance through direct physiological means, warming has been

shown to have substantial effects on biocrust species composition, including macroscopic
components such as moss and lichens (Ferrenberg et al., 2015; Escolar et al., 2012; Maestre et
al., 2015) and microbial communities (Steven et al., 2015; Johnson et al., 2012). Climate models
predict rapidly rising temperatures for already hot and moisture-limited dryland regions,
including the site of our study in the southwestern United States (Stocker, 2014; Jardine et al.,
2013). Forecasts of future precipitation patterns are less certain, but overall drier conditions with
changes in precipitation event size and frequency are likely (Seager et al., 2007). Climate models
predict increases in dryland annual average temperature of up to 4 °C by the end of the 21$^{st}$
century, as well as significant alterations to the amount and timing of rainfall (Christensen et al.,
2007). For example, the Intergovernmental Panel on Climate Change (IPCC) A1B scenario
suggests a decrease in precipitation amount of 5-10% for the southwestern U.S., as well as
significant changes to the timing and magnitude of precipitation (D'Odorico and Bhattachan,
2012). Across many ecosystems, including drylands, both plant C uptake and soil respiration
show an optimum, such that rates are positively correlated with increased temperatures and
moisture (Wu et al., 2011) until a point at which high temperatures (often accompanied with
drying) begin suppressing both photosynthesis (e.g., Wertin et al., 2015) and soil respiration
(Tucker and Reed, 2016). Drought also tends to reduce vascular plant production and respiration,
with greater sensitivity in drier areas (Knapp et al., 2015). In soils overlain by biocrusts
(hereafter, biocrusted soils) specifically, temperature and moisture are key physiological
parameters for C flux (Grote et al., 2010; Darrouzet-Nardi et al., 2015) and, although few, the
warming experiments that do exist suggest that biocrusted soils will have higher net $CO_2$ efflux
with a warming climate (Darrouzet-Nardi et al., 2015; Maestre et al., 2013). There is evidence
for a limit to this association though, with very high temperatures leading to reduced biotic
activity, including microbial respiration, in biocrusted soils (Tucker and Reed, 2016).

To improve our understanding of dryland C flux responses to global change, we used a

warming by watering manipulation experiment on the Colorado Plateau established in 2005.
When the study began, we explored the hypothesis that warming would increase net losses of
$CO_2$ from soils covered with late successional biocrusts (~50% moss, ~30% lichen cover) via
detrimental impacts on biocrust physiology caused by warming. At the same time we wanted to
explore how altered precipitation could directly affect biocrust soil $CO_2$ exchange and/or interact
with the effects of increased temperatures. These early results supported the basic hypothesis
concerning the warming-only treatment, showing that warming led to increased $CO_2$ loss after 1-
2 years, with the largest differences during periods in which soils were wet enough to support
substantial biocrust photosynthesis (Darrouzet-Nardi et al., 2015). Crucially, we also found that
the increased frequency of small frequent precipitation events negatively affected biocrusts: the
treatment caused the death of a major biocrust component, the moss *Syntrichia caninervis* (Coe
et al., 2012; Reed et al., 2012; Zelikova et al., 2012). This finding represented a substantial
alteration to the system and led to a second phase of the experiment. In this phase, we ceased the
watering treatment that had caused moss death and increased the warming treatment from 2° to
4° to see if greater warming would negatively impact biocrusts. We found that the greater
warming did in fact reduce moss and lichen cover as well, though not as rapidly as the watering
treatment (Ferrenberg et al., 2015). Here we report the C balance response to these multiple
phases of the experiment. Our main goals were to: (1) determine if the increased net soil $CO_2$
loss observed after a year of warming was maintained after 8 years, and (2) to assess how the
altered precipitation patterns affected net soil $CO_2$ exchange during the early phase when mosses
were dying and, then later, after mosses were lost and the increased watering had ceased.

**2 Materials and Methods**

**2.1 Site Description**
The study was located in a semiarid ecosystem on the Colorado Plateau (36.675 N, -109.416
W; elevation = 1310 m; mean annual temperature = 13 °C, mean annual precipitation = 269 mm;
WRCC 2014) that supports multiple species of grasses and shrubs. Soils are Rizno series
Aridisols and the dominant plants include *Achnatherum hymenoides, Pleuraphis jamesii,*
*Atriplex confertifolia,* and *Bromus tectorum.* Biocrust communities are dominated by the
cyanobacterium *Microcoleus vaginatus,* the moss *Syntrichia caninervis,* and the cyanolichens
*Collema tenax* and *Collema coccophorum*. The site is on a moderate hillslope (~10%)
surrounded by steep gullies that make it hard to access for livestock, which may explain its relict
biocrust and plant composition that includes late successional crusts with well-developed
communities of native grasses and shrubs, similar to sites found in Canyonlands National Park
(Belnap and Phillips, 2001). Rainfall during the study period was distributed around the mean
(Table 1, Fig. S1), with several slightly above average years including the first and last year of
the experiment (2006: 294 mm; 2014: 304 mm), and one year with substantial drought (2012:
122 mm). Rainfall and temperatures went up and down across years, with no notable directional
shift over the 9-year course of the study. Long-term records from a nearby weather station in
Moab, UT show that mean annual temperatures have been increasing (21.3 °C for 1900-1924 vs.
22.9 °C for 1991-2016, a difference of 1.5 °C). Precipitation trends since 1925 do not show a
clear trend (Fig. S2).

**2.2 Warming and watering treatments**
The experiment contained 20 plots with 5 replicates ($n = 5$) for each of 4 treatments: *control*,
*warmed*, *watered*, and *combined* (warmed + watered). Plots were $2 \times 2.5$ m in size and grouped
into 5 blocks determined by spatial location on the hillslope. Each plot contained one automated
$CO_2$ chamber (described below). The warming treatment began in October 2005 in plots fitted
with 800 W infrared radiant (IR) heat lamps (Kalglo Model MRM-2408) mounted at a height of
1.3 m. Control plots had dummy lamps that do not provide heat. The heating treatment was
regulated by altering the voltage supplied to each lamp. While some drying of soil moisture from
the lamps may have occurred, we saw little evidence for this phenomenon in soil moisture
values, with drying after precipitation events occurring at similar rates in all treatments (Fig. S3).
A previously published analysis also reported no easily detectable moisture effects from the
infrared lamps in either this experiment or a similar co-located experiment despite soil moisture
probes at 2, 5, and 10 cm throughout all plots (Wertin et al., 2015). However, we cannot rule out
very shallow surface moisture effects, which could be important (Tucker et al., 2017).
The target temperature increase was ambient soil temperature +2 °C from 2005-2008, at
which point a second lamp was added to each plot and the warming treatment was increased to
+4 °C where it remained through the end of the automated chamber sampling in September of
2014. The treatment temperatures were increased from 2 to 4 °C above ambient in order to better
match changing predictions of future temperature by 2100 (Christensen et al., 2007). To simulate
predictions of increased frequency of small precipitation events (Weltzin et al., 2003;
Christensen et al., 2007), water was added in 1.2 mm events manually with backpack sprayers
and was applied 40 times from May 31-Sep 20, 2006 and 36 times from June 14-Sep 20 in 2007,
with an average time between watering of 2.8 days (~4x natural frequency; Table 1). This
watering treatment continued through 2012 (Table 1). The amount of water varied by year
because watering did not occur on days when natural rainfall occurred. Watering was stopped
after 2012 because the late successional biocrust community had been eliminated after the first
year and was showing no further change through time (Reed et al., 2012; Ferrenberg et al.,
2015).

**2.3 Net soil exchange measurements with automated chambers**
Carbon dioxide fluxes were assessed with automated $CO_2$ flux chambers, described in detail
in Darrouzet-Nardi et al. (2015). The chambers were placed within the soil, open at the bottom
and have clear lids at the top that are closed once per hour for 3 min to assess net $CO_2$ flux. The
chambers allow in sunlight and hence allow photosynthesis by biocrust organisms. Fluxes of
$CO_2$ during that time are calculated as the rate of change in $CO_2$ concentrations during the 3 min
period. During that 3 min period, $CO_2$ was recorded every 2 s and averaged every 10 s. Aberrant
points were down-weighted with a smoothing function ('supsmu' implemented in MATLAB;
Friedman, 1984), allowing a robust calculation of slope for a given 3-minute interval (Bowling et
al., 2011). The chambers were 30 cm tall $\times$ 38 cm inner diameter, covering a soil surface area of
0.11 $m^2$. Chambers were installed to a depth of 27 cm in the soil, leaving ~3 cm of the chamber
protruding above the soil surface. The chambers were placed in plot locations containing
biocrusts but no vascular plants. Values from these chambers were reported as net soil exchange
(NSE) of $CO_2$. The concept of NSE is defined in Darrouzet-Nardi et al. (2015) to include
biocrust photosynthesis as the sole form of $CO_2$ uptake (i.e., because the chambers do not include
vascular plants) along with $CO_2$ losses via respiration from biocrusts, other soil microbes, plant
roots, and any abiotic soil sources. While it would have been ideal to operate the chambers year
round for the entire course of the experiment, it was beyond the operational capacity of the
project to do so and there are times when the systems were not operational. The chambers have
more frequent malfunctions during the winter due to weather conditions, so those months are
least represented. There were intermittent automated chamber measurements in 2012, the last
year of watering, crossed with the higher warming level, providing enough data for analyses of
daily patterns, though not enough to assess seasonal total rates.
Biocrust community composition of the autochambers was measured at the initiation of the
experiment in 2005 and again in 2017. Assessment of the biocrust community was performed
using a frame that covered the autochamber area in which the cover of thirty-one individual 25.8
$cm^2$ squares as estimated for all biocrust species. The total cover of each species was summed
from the individual quadrats and the quadrats covered 800 $cm^2$ of the chambers' 1100 $cm^2$ area.
**2.4 Imputation and statistical analysis**
Hourly data from the automated chambers were collected from January 1, 2006 - September
20, 2007, February 19 - November 17, 2013, and February 14 - November 17, 2014, for a total of
28,058 time points for each of the 20 chambers. Of these time points, 29% of the data were
missing, primarily due to technical issues with the chambers. To allow calculations of
cumulative NSE, data were imputed following the same procedure as in our previous work
(Darrouzet-Nardi et al., 2015). Data were assembled into a data frame containing columns for (i)
each of the 20 chambers; (ii) environmental data including soil and air temperature, soil
moisture, 24-hour rainfall totals, photosynthetically active radiation (PAR); and (iii) six days of
time-shifted fluxes (before and after each measurement; i.e., -72 h, -48 h, -24 h, +24 h, +48 h,
+72h) for one chamber from each treatment, soil temperature, and soil moisture. Lagged values
were added due to their ability to greatly improve prediction of missing time points, particularly
for short time intervals such as those caused by, for example, several hours of power outage at
the site. One data frame was created for each of the three continuous recording periods: 2006-
2007, 2013, and 2014 and each was imputed separately. Imputation was performed using the
missForest algorithm, which iteratively fills missing data in all columns of a data frame using
predictions based on random forest models (Stekhoven and Buhlmann, 2012; Breiman, 2001).
After imputing the hourly values, cumulative fluxes were calculated by summing NSE over
seven-month periods (February 19 - September 19) for each year (2006, 2007, 2013, and 2014).
This seven-month period was selected due to availability of data in all four analysis years. The
total number of cumulative fluxes evaluated was 80 (4 years $\times$ 4 treatments $\times$ 5 replicates). We
also made separate cumulative estimates of time periods in which we observed active
photosynthesis, defining these periods as days during which the NSE values were -0.2 $\mu$mol $CO_2$
m$^{-2}$ s$^{-1}$ or lower, with more negative numbers showing higher net photosynthesis. These periods
typically correspond to times with sufficient precipitation to activate biocrusts. The effect of the
*warmed*, *watered*, and *combined* treatments on cumulative NSE values were evaluated by
calculating the size of the differences between each treatment and the control (Nakagawa and
Cuthill, 2007; Cumming, 2013). Treatment differences, which we notate as $t_d$, were calculated as
treatment – control (paired by block) with 95% confidence intervals estimated using mixed
effects linear models for each year with treatment as a fixed effect and block as random effect
(Pinheiro and Bates, 2000). Analyses were facilitated by a custom-made R package "treateffect",
available at https://github.com/anthonydn/treateffect. The data used for these analyses are
available at https://doi.org/10.6084/m9.figshare.6347741.v1. Finally, to evaluate differences over
time, differences between 2006 data for each treatment and each subsequent year were
calculated, also using mixed effects models.

**3 Results**

Biocrust cover within the soil collars used by the automated chambers was relatively similar

in all treatments at the beginning of the experiment, with an average of 49% moss and 31%
lichen in each treatment (Fig. 1). Between 2005 and 2017, these percentages fell in all treatments
including the controls, eventually being replaced primarily by lightly-pigmented cyanobacterial
crusts, probably *Microcoleus vaginatus* (Gundlapally and Garcia-Pichel, 2006). Lichen went to
<3% in all treatments. Mosses were more variable, remaining at 25% in controls, but falling to
7% in warmed plots and to 0% in both watering plots. Cyanobacteria cover started at 0% in all
chambers and rose to 50-90%.

Seasonal time courses of NSE showed similar patterns among years and treatments, with

peaks in NSE in the spring associated with peak vascular plant activity, and peaks in both
negative and positive NSE associated with rain events (Fig. 2a). In the early time period (1-2
years after treatments began), the supplemental 1.2 mm watering treatment caused large "puffs"
of $CO_2$ when water was added. By the final year of watering (2012), the size of these puffs was
substantially smaller and after watering ceased (2014), they did not occur even with natural
rainfall events (Fig. 3).

In the early time period (2006-2007), interannual comparisons of cumulative Feb. 19 - Sep.

19 (seven-month) $CO_2$ fluxes were consistent with the hypothesized trend of the warming and
watering treatments increasing $CO_2$ flux to the atmosphere. In the early time period, shortly after
the establishment of the treatments, we observed higher NSE (greater movement of $CO_2$ from
soil to the atmosphere) in both watered and combined treatment plots, with less evidence of
difference in the warming only treatment (Fig. 4a; Table 2). Fluxes were similar between 2006
and 2007 (Table S1).
In the later time period (2013-2014), the treatments showed varying results. In 2013, after the
watering treatment had ceased, we observed a reversal of the treatment trend from the early
period, with lower $CO_2$ efflux from soils in all three treatments (Fig. 4a; Table 2). This trend was
particularly visible in the months of May and June (Fig. 2a,b). However, in the following year,
2014, a wet year with high spring rainfall (Table 1, Fig. 2a), all plots showed the highest $CO_2$
efflux observed in the experiment (e.g., 36.2 [21.7, 52.9] $\mu$mol m$^{-2}$ s$^{-1}$ higher compared to 2006
in control plots; Table S1). While no obvious treatment effects were observed, treatment effect
sizes were relatively poorly constrained due to the higher variation that year (Table 2).
Interannual comparisons of cumulative $CO_2$ fluxes during periods of active photosynthesis
showed higher photosynthesis in all treatments during the early measurement period (e.g., 2006
warmed $t_d = 4.1$ [-0.1, 8.2]; Fig. 4b; Table 2). In the later period (8-9 years after treatments
began), subsequent to the cessation of watering, warmed plots still showed elevated $CO_2$ losses
during periods of active photosynthesis but this difference was smaller than in the earlier
measurements (e.g., 2013 warmed $t_d = 1.3$ [-0.5, 3.1]; Fig. 4b; Table 2). In contrast, watered plots
that were not warmed were similar to control plots.
In examining the daily cycles in the hourly data, further detail on the nature of the treatment
effects was observed. After one year, watered treatments in which mosses had died showed
strong reductions in $CO_2$ uptake capacity during wet-up events, but warmed treatments still
showed a similar maximum uptake capacity relative to controls (e.g., minimum NSE on October
15, 2006 control = -0.93 $\pm$ 0.19 $\mu$mol m$^2$ s$^{-1}$; warmed = -0.89 $\pm$ 0.11, watered = -0.35 $\pm$ 0.06,
combined = -0.2 $\pm$ 0.08; Fig. 5a). However, after 8 years of treatment, clear differences were
present in the $CO_2$ flux dynamics in response to natural rainfall events (Fig. 5b). Biocrusted soils
in control plots still exhibited substantial net uptake of $CO_2$ (e.g., minimum NSE on August 14,
control = -0.68 $\pm$ 0.12 $\mu$mol m$^2$ s$^{-1}$), whereas the other treatments showed less uptake relative to
the control, with a similar trend visible on August 23rd.

## 4 Discussion

### 4.1 Early period: 2 °C warming × watering (2006-2007)

The increase in $CO_2$ effluxes in the watered treatments during the early period (Fig. 4, Table 2) were likely driven by both the loss of photosynthetic biocrust organisms during that time (Reed et al., 2012), as well as increased soil respiration from soil heterotrophs. Moss death may have contributed to net soil C loss via (i) eliminating $CO_2$ uptake from this important biocrust $CO_2$-fixer (Reed et al., 2012; Coe et al., 2012); and (ii) decomposition of dead mosses. Elevated soil respiration with warming and watering is broadly consistent with the results of similar experiments across many ecosystems (Wu et al., 2011; Rustad et al., 2001), dryland sites specifically (Nielsen and Ball, 2015; López-Ballesteros et al., 2016; Patrick et al., 2007; Thomey et al., 2011), and previously documented effects in biocrusted soils at this site and others (Darrouzet-Nardi et al., 2015; Maestre et al., 2013; Escolar et al., 2015). In the warmed treatment, elevated NSE was not as evident in 2006 as in the watered and combined treatments, and this is consistent with the biocrust community changes. While moss died off quickly in the watered plots, mosses in the warmed plots took longer to show negative effects (Ferrenberg et al., 2017). Indeed, increased $CO_2$ efflux with warming was clearer in the following year (2007) and moss cover was substantially reduced by 2010 (Ferrenberg et al., 2015). Such rapid species composition changes have been repeatedly implicated as drivers of system change in drylands, even with seemingly subtle changes in climate (Wu et al., 2012; Collins et al., 2010).

### 4.2 Late period: 9 years warming (2-4 °C) × legacy watering (2013-2014)

During the later period (2013) when warming had been increased to +4 °C (in 2009) and watering had ceased (effectively making the treatments: control, +4 °C, legacy watering, and +4 °C × legacy watering), several differences in treatment effects emerged in comparison to the early measurement period (2006-2007). First, the trend in the 2013 seven-month cumulative $CO_2$ fluxes (Fig. 4, Table 2) were reversed from those of the early measurement period (2006-2007), with the control plots having the highest NSE and all other treatments showing lower $CO_2$ efflux. The reversal of the NSE trend in the +4 °C and +4 °C × legacy watering treatments is likely influenced by changes in biocrust community composition, with mosses largely eliminated in relation to the control plots where about half of the mosses were retained (Fig. 1). By 2013,

lower NSE in warmed and watered plots may have been linked to the completion of moss and
lichen decline and thus cessation of fluxes from sources such as decomposition or exudation.
Reductions in biocrust cover were also observed in the control plots perhaps due to the longer-
term effects of infrastructure, human variation in community assessment, or natural variation in
community composition (Belnap et al., 2006), and such changes could help explain the higher
NSE in controls in 2013. Another possibility is that the reduced vascular plant photosynthesis
observed for multiple plant species with warming in this area (Wertin et al., 2015; Wertin et al.,
2017) reduced plant allocation of C belowground. This trend could reduce root C efflux and
heterotrophic breakdown of root exudate C, leading to the observed lower NSE values. A
number of warming experiments in more mesic systems that do not have photosynthetic soils
have shown an initial warming-induced increase in soil $CO_2$ respiratory loss followed by
subsequent declines in warmed plots; in these situations, reduced soil C availability for
heterotrophic respiration and changes to heterotroph C use efficiency are often suggested to play
a role (Bradford et al., 2008; Bradford, 2013; Tucker et al., 2013). Such effects would also be
consistent with drying from the infrared heat lamps, a mechanism that was supported in a
Wyoming grassland experiment (Pendall et al., 2013). Our soil moisture data showed little
evidence of such drying effects (Fig. S3). However, with a minimum moisture probe depth of 2
cm, we may have missed moisture effects relevant only to the top several millimeters of soil, an
area of current active investigation at the site: more recent results suggest that surface moisture
(0-2 mm) can be a potent predictor of soil C fluxes on these biocrusted soils (Tucker et al.,
2017). The reduction in $CO_2$ efflux with warming was also seen in a nearby set of plots in 2011,
in which soil respiration was measured at individual time points with non-automated chambers
(Wertin et al., 2017). In that study, the reduction with warming was observed three years after +2
°C warming treatment was implemented. The dark respiration measurements were made in the
spring (at peak plant activity) and it was at the same point in the season (see Fig. 2) that we saw
the strongest seasonal driver for the seven-month cumulative data. In sum, although our NSE
data don't allow us to disentangle the driving mechanisms, changes in (i) biocrust composition,
(ii) nearby plant activity, and (iii) possibly surface moisture could all have contributed to the
reversal in the effect of the warming treatment in the late period of the study. Regardless of the
cause, these data suggest large, sustained changes to dryland soil C cycling at our site in response
to climate change treatments.
We also observed reduced NSE values in the 2012-2013 sampling period in plots that were
previously watered plots compared to the control plots, suggesting some legacy treatment effects.
This was likely linked to loss of mosses, cyanobacteria, or changes in vascular plant physiology.
For example, at a European site, biocrusted soil microsites were shown to be a dominant source
of midday soil respiration (Castillo-Monroy et al., 2011). Furthermore, reductions in the
autotrophic biomass seen with the climate treatments could reduce respiration rates (Ferrenberg
et al., 2017; Reed et al., 2016). Plants accustomed to the extra water may also have responded
negatively to its absence, causing reduced physiological activity and hence lower root
respiration, an effect that has been documented in drought simulation experiments (Talmon et al.,
2011). Soil heterotrophs can also show legacy effects of their species composition in response to
changes in precipitation regime (Kaisermann et al., 2017). Water retention may also have been
reduced due to the decline in biocrust cover, an effect for which there is some evidence,
particularly in semiarid ecosystems like our study site (Belnap, 2006; Chamizo et al., 2012).
Mosses have unique adaptations allowing them to absorb high fractions of precipitation without
loss to splash and evaporation (Pan et al., 2016), a process that would be lessened in the climate
manipulation plots due to moss death. In addition to effects on soil moisture, changes in biocrust
community composition can have significant effects on soil nutrient availability (Reed et al.,
2012) and nutrient availability can be tightly coupled with soil respiration rates (Reed et al.,
2011). Although the NSE data do not allow us to determine which gross C fluxes caused the
opposing treatment effects between the early (2006-2007) and late (2012-2013) measurement
periods, the observation of a reversal like this is important because if the larger $CO_2$ loss had
been sustained, it would have indicated the potential for large feedbacks to increasing
atmospheric $CO_2$ concentrations.
Interestingly, the $CO_2$ loss reversal observed in 2013 did not continue in 2014, likely due to
the higher rainfall, particularly during spring. In 2014, we saw high NSE in all plots in the seven-
month cumulative data, with no significant differences among treatments. Accompanying the
higher precipitation in 2014 – which occurred in a series of large rain events in April and May –
perennial plants were noticeably greener and there was a flush of annual plants (S.C. Reed,
*unpublished data*). During wet conditions, warmed plots had higher NSE values, which could
have been due to higher root respiration or higher subsoil microbial activity, potentially linked to
root turnover or rhizodeposition (Jones et al., 2004). These results from the later period of the
experiment (2013-2014) underscore that taking a long-term perspective (i.e., nearly a decade of
warming) may be necessary for understanding climate change effects, particularly those that
maintain interactions with species composition changes. Further, these data suggest more
complexity in soil $CO_2$ efflux controls, such that some systems may not manifest a simple
transition from temperature-induced increases in soil $CO_2$ loss to temperature-induced decreases
at later stages of warming. The interannual variations in the magnitude of NSE fit with results
from other drylands that show high interannual variation in net ecosystem exchange (NEE) as
measured with eddy flux towers (Biederman et al., 2017). At least one other longer-term
manipulation in a dryland has also observed early stimulation of plant growth with warming that
then lessened over time, with longer-term effects driven by changes in species composition (Wu
et al., 2012). The finding that decadal-scale studies can have mixed and context-dependent
effects not visible at the annual scale (Nielsen and Ball, 2015) is exemplified in our study by the
reversal in effects seen in 2013, followed by the swamping out of those effects in a subsequent
wet year.

**4.3 Source of $CO_2$ efflux**

Observed NSE fluxes were almost always net positive (C loss to atmosphere), indicating that

soil profile C losses are greatly outpacing biocrust photosynthetic uptake (Fig. 2). This
necessitates a non-biocrust C source as biocrusts cannot persist with consistently negative C
balance (e.g., Coe et al. 2012). The $CO_2$ efflux data also support these non-biocrust sources. For
example, though we did lose biocrusts, even in control plots, C losses continued even in plots
where the larger biocrust constituents were gone (e.g., watered plots in 2014). Besides biocrust
organisms, there are three other potential sources of $CO_2$ efflux: soil heterotrophs, vascular plant
roots, and pedogenic carbonates (Darrouzet-Nardi et al., 2015). All three are possible
contributors and further work is needed to partition their contributions.

We would expect the biocrusts themselves to have the biggest impact on NSE when soils are

wet and biocrusts are active. During such time periods, we saw treatment effects that were
distinct from the seven-month totals (Fig 2b), which could be interpreted as evidence of a
biocrust signal that did not follow the general vascular plant trends of spring activity. Indeed,
several pieces of evidence point directly to a biocrust signal. First, in the later time period (2013-
2014), the reduction in minimum daily NSE during precipitation events (Fig. 5) suggests that
loss of biocrust $CO_2$ uptake contributed to higher net C loss from these soils. In particular, the
*combined* treatment lost a large proportion of its capacity to assimilate C, as well as much of the
biocrust biomass. Second, the decline in the size of the "puffs" of $CO_2$ that were associated with
the 1.2 mm watering treatments are likely driven by declines in biocrust activity (Fig. 3), as these
small watering events primarily affect the surface of the soil. These biocrust activities could
include both biocrust respiration and decomposition of dead biocrust material. In our previous
work (Darrouzet-Nardi et al., 2015), we saw evidence of these puffs in control plots without
supplemental watering, though they were presumably not frequent enough to kill the mosses
under natural conditions, a situation that could be altered if precipitation is altered in the future
(Reed et al., 2012; Coe et al., 2012).

Heterotrophic respiration could also be a substantial contributor to the $CO_2$ effluxes we

observed. The soil $CO_2$ efflux was observed rapidly after each rain pulse (natural or
experimental), which could indicate soil heterotrophic respiration since plant photosynthesis may
take longer to become activated (López-Ballesteros et al., 2016). The soil organic C pool in these
soils includes ~300 g C m$^{-2}$ in the 0-2 cm biocrust layer, which would be depleted rapidly if it
were the sole C source. However, the sub-biocrust 2-10 cm layer has ~430 g m$^{-2}$ and soils are on
average 50 cm deep at the site, suggesting that the total sub-crust soil C is >1500 g C m$^{-2}$ (data
not shown). With a C pool of that magnitude, depletion of soil organic matter C stocks could be
substantial contributors to the C losses we observed. However, if losses on the order of 62 g C m$^{-}$
$^{2}$ (the amount lost in control plots during 2006) were to continue, these stocks would be
completely depleted (which normally does not occur in soils) in ~25 years, suggesting another
source is also extremely likely.

Root respiration is a contributor we consider highly likely. During excavations of the

chambers in 2017, root biomass was observed inside the chambers, making a root signal
plausible. Previously published measurements from a nearby site that did not have a well-
developed biocrust community showed tightly coupled measurements of plant photosynthesis
with soil respiration directly beneath plant canopies (Wertin et al., 2015) while correlations
between soil C concentration and soil respiration were much weaker (Wertin et al., 2018).
Furthermore, the seasonal NSE trends are broadly consistent with a plant photosynthetic signal,
particularly the peak in fluxes during the spring growing season, which coincides with plant
uptake as indicated by negative NEE seen using eddy flux towers (Darrouzet-Nardi et al., 2015;
Bowling et al., 2010). The interannual trends presented in this study are also consistent with a
plant signal: for example, the wettest year, 2014, was the year in which the highest $CO_2$ efflux
rates were observed, a phenomenon that was likely driven by both increased activity in
perennials and the flush of annual plants observed in that year. Finally, not only is a strong plant
signal likely in these NSE measurements, but the interpretation of the treatment differences,
particularly the unexpected finding of a reversal in the seven-month cumulative fluxes discussed
above, is clearer in light of a plant signal. We believe that by 2013, reductions in plant
productivity could have resulted in reduced root respiration in the non-control plots.
Finally, pedogenic carbonates can contribute to $CO_2$ efflux and we cannot rule out their
contribution in this study (Emmerich, 2003; Stevenson and Verburg, 2006). Some studies
suggest that $CO_2$ efflux during dry periods is likely to be from inorganic sources (Emmerich,
2003). Others make the case that the timing of $CO_2$ efflux from $CaCO_3$ would be more likely to
overlap with the times when plants were active and calcite could be dissolved in conjunction
with a source of acidity such as acid deposition, root exudation, or nitrification (Tamir et al.,
2011). Either way, long-term loss of $CO_2$ from dissolved calcite from our site cannot be ruled out
and a field investigation of the isotopic composition of released $CO_2$ would be particularly
valuable in assessing inorganic contributions.

**4.4 Conclusions**
Both warming and watering with the associated moss death initially led to higher $CO_2$ losses
in our experimental plots. After the cessation of watering, the patterns in the C balances were
reversed in an average moisture year (2013), with the climate manipulation plots of all
treatments showing lowered soil $CO_2$ loss relative to controls. These data are in line with
warming experiments from a range of climates suggesting warming-induced increases in soil
$CO_2$ are not a long-term phenomenon, at least within these experimental frameworks. Moreover,
in a subsequent wet year (2014), $CO_2$ fluxes were uniformly high among treatments. When
focusing just on periods of active biocrust photosynthesis, after 8 years, biocrust photosynthetic
performance was much weaker in both warmed and legacy watered treatments relative to the
control plots despite biocrust changes in control plots as well. These results suggest that the
community composition changes that are highly likely in dryland plants (Collins et al., 2010; Wu
et al., 2011) and biocrusts (Ferrenberg et al., 2017; Johnson et al., 2012) as a response to global
change are likely to affect C balances even if effects are not consistent year to year. Our results
show how community shifts, such as the loss of a major photosynthetic component like mosses,
will contribute to an altered C balance of these biocrusted soils. Finally, our results underscore a
strong role for biocrust, root, and possibly soil heterotrophic and inorganic signals in NSE,
suggesting that further study of the balance of plant assimilation and root/rhizosphere respiration
of C, as well as patterns in biocrust C, in response to climate change will be an important
determinant of future C fluxes in drylands.

*Author Contributions.* J.B. initiated the experiment and J.B. and S.C.R. gained funding for the
work. All authors performed the experiment, with E.E.G. leading design and construction of the
automated chambers and data management. A.D.N analyzed the data and led manuscript writing,
and all authors contributed to the writing.

*Acknowledgements.* This work was supported by the U.S. Department of Energy Office of
Science, Office of Biological and Environmental Research Terrestrial Ecosystem Sciences
Program, under Award Number DE-SC-0008168, as well as US Geological Survey's Climate
and Land Use and Ecosystems Mission Areas. We thank the multitudinous technicians and
collaborators who have contributed to the field operations on this project. Any trade, product, or
firm name is used for descriptive purposes only and does not imply endorsement by the U.S.
Government.

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

| Year | Moab MAT (°C) | Moab MAP (mm) | Study site MAT (°C) | Study site MAP (mm) | Spring precipitation (mm) | Supplemental water (mm) | First watering date | Last watering date | Number of watering days |
|---|---|---|---|---|---|---|---|---|---|
| 2006 | 22.6 (2) | 208 (0) | 21.4 (0) | 294 (0) | 22 | 48 | May 31 | Sep 20 | 40 |
| 2007 | 22.9 (8) | 191 (4) | 22.1 (0) | 223 (0) | 68 | 42 | Jun 14 | Sep 20 | 36 |
| 2008 | 21.8 (4) | 138 (0) | 22.6 (0) | 200 (0) | 62 | 44.4 | Jun 17 | Sep 23 | 43 |
| 2009 | 21.9 (1) | 126 (0) | 20.8 (1) | 189 (0) | 57 | 27.8 | Jun 10 | Sep 04 | 32 |
| 2010 | 21.4 (0) | 204 (0) | 20.0 (13) | 286 (13) | 51 | 48 | Jun 09 | Sep 29 | 40 |
| 2011 | 21.7 (0) | 161 (0) | 20.0 (1) | 199 (0) | 71 | 42 | Jun 13 | Sep 19 | 36 |
| 2012 | 23.6 (1) | 92 (1) | 22.1 (85) | 122 (84) | 9 | 54 | Jun 04 | Oct 05 | 45 |
| 2013 | 20.7 (2) | 183 (2) | 19.3 (36) | 253 (32) | 43 | 0 | May 31 | Sep 20 | 0 |
| 2014 | 22.8 (0) | 208 (0) | 21.5 (1) | 304 (0) | 73 | 0 | Jun 14 | Sep 20 | 0 |


Table 1. MAT = mean annual temperature. Values are shown for the nearby Moab site (see Fig.
S2 for long-term record) as well as for the instruments at our study site. Values in parentheses
indicate the number of days of missing data for the given year. MAP = mean annual precipitation
and spring precipitation totals were determined by a rain gauge at the study site. Detailed timing
of temperature and precipitation over the study period are shown in Fig. S1. Supplemental water
was only added to the watering and combined treatments and was not added on days when
natural precipitation occurred. Spring rainfall is from day of year 80-173 and is the time of peak
plant growth.

| Year | Comparison | Seven-month periods $t_d$ (g C m$^{-2}$) | Active photosynthesis periods $t_d$ (g C m$^{-2}$) |
|---|---|---|---|
| 2006 | Warmed - Control | 5.1 [-9.7, 19.9] | 4.1 [-0.1, 8.2] |
| 2006 | Watered - Control | 14.6 [-0.2, 29.4] | 5 [0.8, 9.1] |
| 2006 | Combined - Control | 9.8 [-5.1, 24.6] | 7.6 [3.5, 11.8] |
| 2007 | Warmed - Control | 6.1 [-6.7, 18.7] | 2 [0.6, 3.5] |
| 2007 | Watered - Control | 10.9 [-1.8, 23.6] | 1.5 [0, 2.9] |
| 2007 | Combined - Control | 8.33 [-4.4, 21.0] | 2.6 [1.2, 4.1] |
| 2013 | Warmed - Control | -10.7 [-27.7, 6.2] | 1.3 [-0.5, 3.1] |
| 2013 | Watered - Control | -15.3 [-32.2, 1.6] | -0.1 [-1.8, 1.7] |
| 2013 | Combined - Control | -11.8 [-28.7, 5.2] | 0.9 [-0.9, 2.7] |
| 2014 | Warmed - Control | -1.2 [-30.6, 28.1] | 2.9 [-1.1, 7] |
| 2014 | Watered - Control | -4.0 [-33.3, 25.3] | 0.4 [-3.7, 4.4] |
| 2014 | Combined - Control | -6.2 [-35.5, 23.1] | 1.6 [-2.4, 5.6] |



Table 2. Effect sizes of our treatments are shown as mean differences in NSE between treatments
and controls with 95% confidence intervals ($t_d$). Values were calculated as the control plot rate
subtracted from the rate in the treatment plot, with positive values indicating higher NSE values
in the treatment plot relative to the control and vice versa. Analyses correspond to the NSE data
shown in Fig. 4. Note that all underlying fluxes are positive (source to atmosphere), but here the
*differences* between treatments are shown.





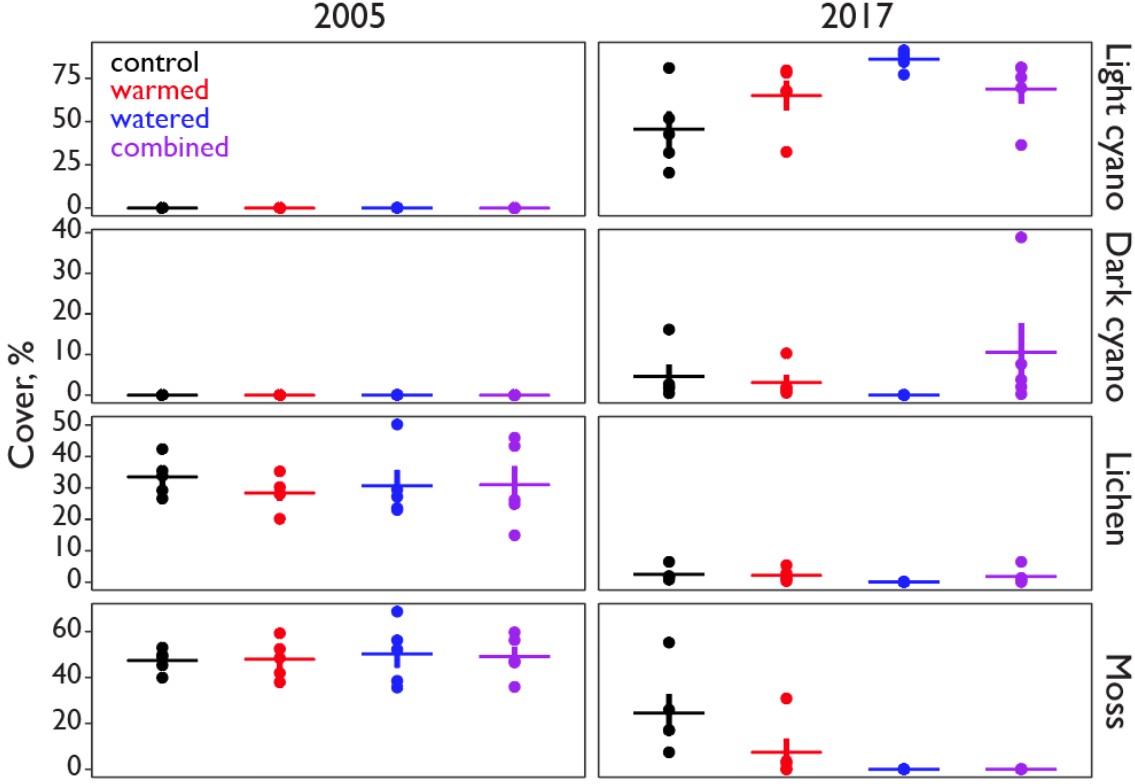



Fig. 1. Cover (%) of major biocrust constituents inside of the automated $CO_2$ flux chambers

representative of the early and later periods of the study.


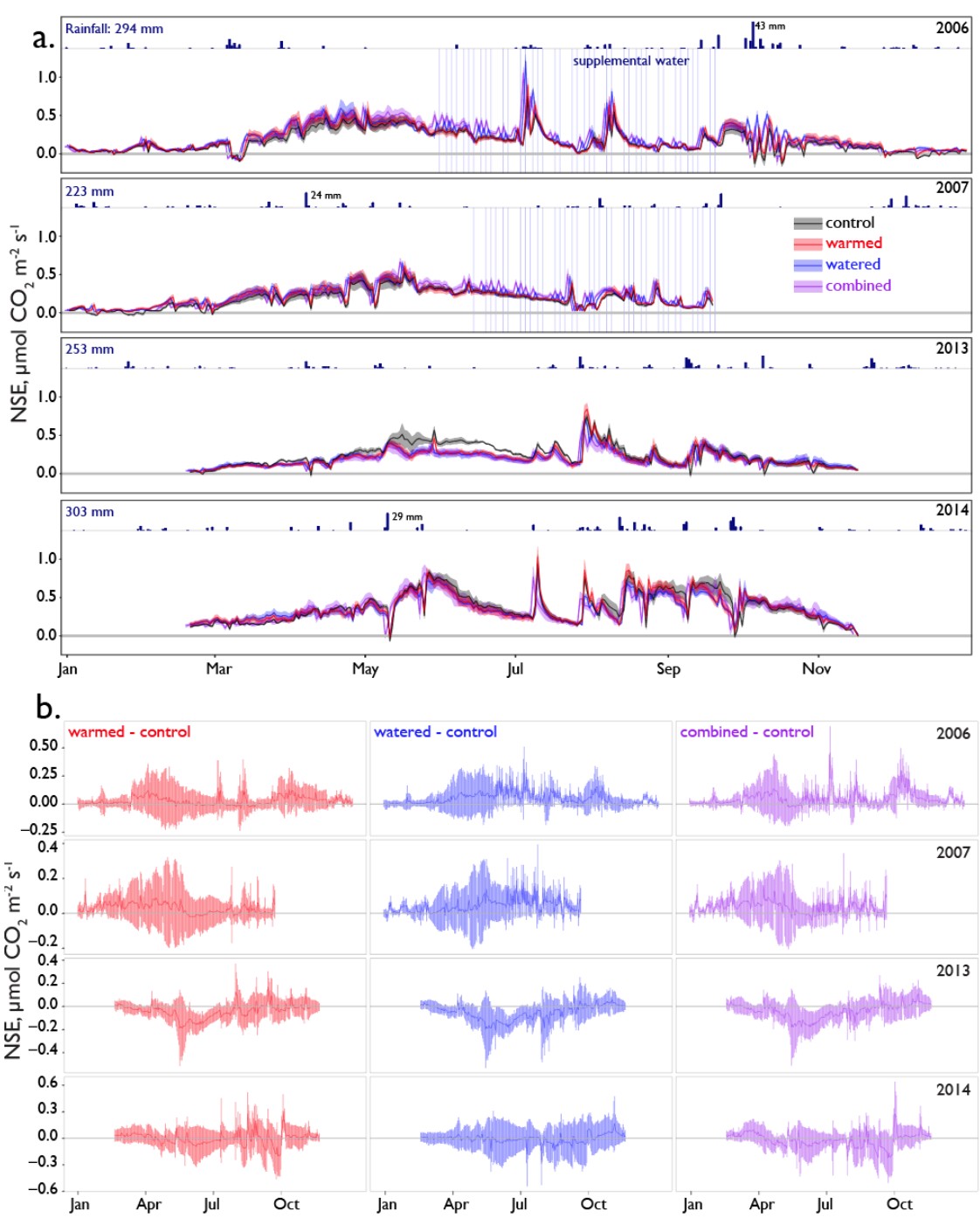


Fig. 2. a. 24-hour average net soil exchange (NSE) of $CO_2$ through all treatments and years.
Dates of supplemental watering applications are shown as vertical blue lines. Ribbons indicate ±
1 SE. Precipitation is shown above each year's data, with annual totals shown on the left and the
size of several of the largest events noted for scale. Means for each treatment are shown with
different colors representing different treatments (control = black, warmed = red, altered
monsoonal precipitation [watered] = blue, warmed × watered [combined] = purple). Positive
NSE rates depict respiratory losses that were greater than $CO_2$ uptake and negative NSE rates
depict C fixation rates that outpaced respiratory losses. b. Differences between treatments and
control ($t_d$) are shown as solid lines ± 95% CI calculated for each daily average shown with
shading. Values were calculated by subtracting the control rates from the treatment (red =
warmed – control; blue = altered monsoonal precipitation [watered] – control; purple = warmed
× watered [combined] – control).

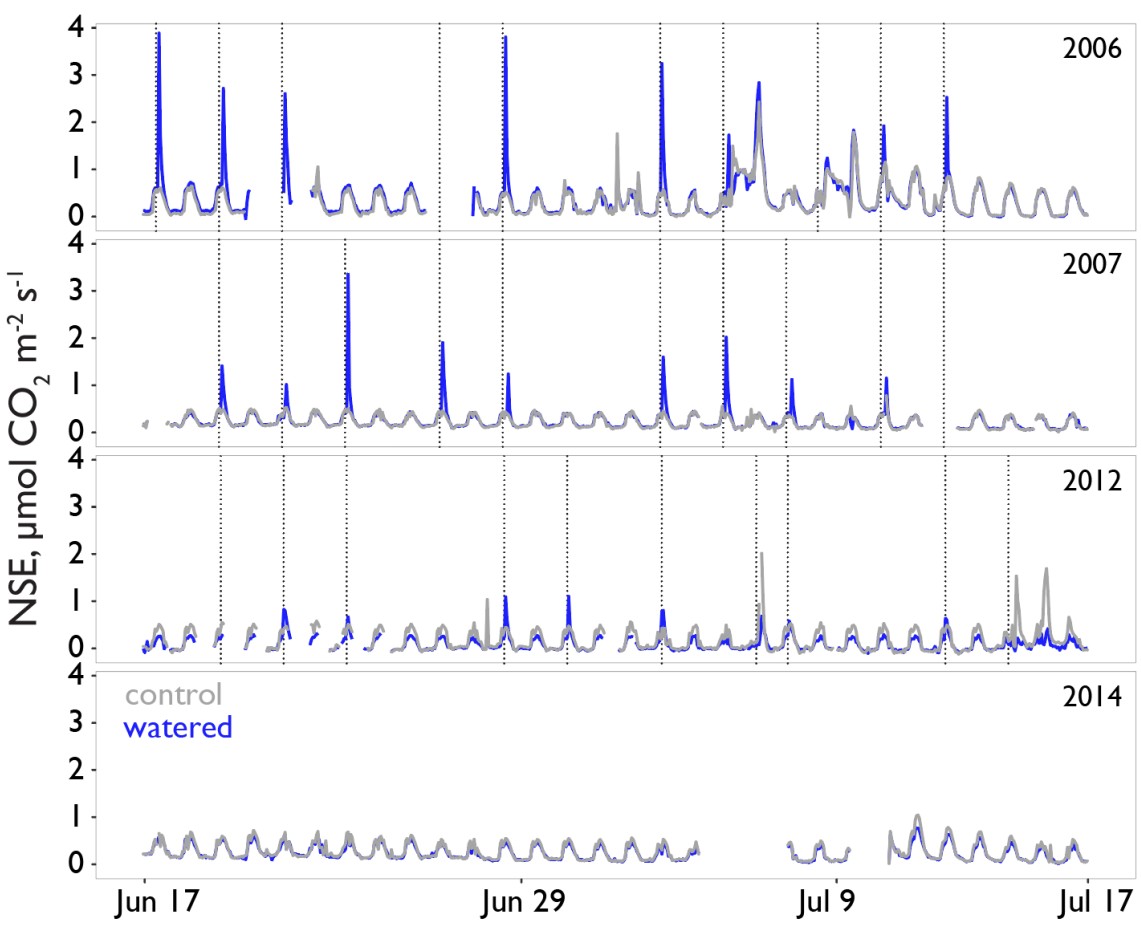


Fig. 3. Interannual comparison of "puffs" of $CO_2$ from single automated flux chambers (watering
treatment, block 2 in blue and comparable control chambers in gray) observed in response to
mid-summer experimental watering treatments. Time resolution is hourly. Plots were
experimentally watered from 2005-2012, with no watering in the final panel (2014). Timing of
the watering treatments is shown by the vertical dotted lines. The puffs shown here are $CO_2$
fluxes at or above ~1 $\mu$mol $CO_2$ m$^2$ s$^{-1}$ and these occurred in response to active watering
treatments.

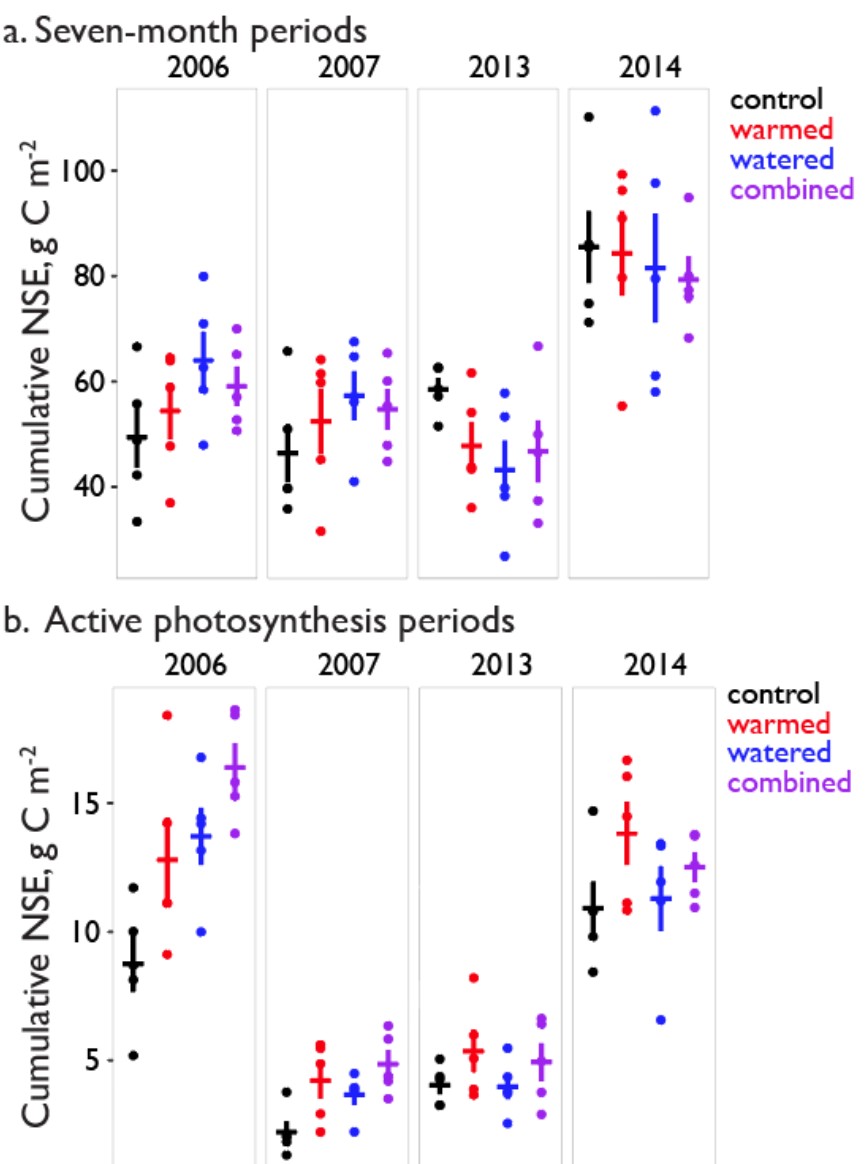


Fig. 4. (a). Seven-month cumulative $CO_2$ fluxes during 4 measurement years: 2006, 2007, 2013, and 2014 for the period of February 19 - September 18, a period chosen due to availability of data in all measurement years. (b) Cumulative $CO_2$ flux during periods with active photosynthesis (defined as days during which NSE was $< -0.2$ µmol $CO_2$ m$^{-2}$ s$^{-1}$ or lower, largely corresponding with wet periods). Though selection was made on this daily minimum, numbers are positive because 24 hour totals during these periods were still largely net sources of $CO_2$ to the atmosphere despite active photosynthesis during peak hours. Dots indicate values from individual automated chambers and horizontal and vertical bars indicate mean $\pm$ SE. For effect sizes associated with each treatment, see Table 2.

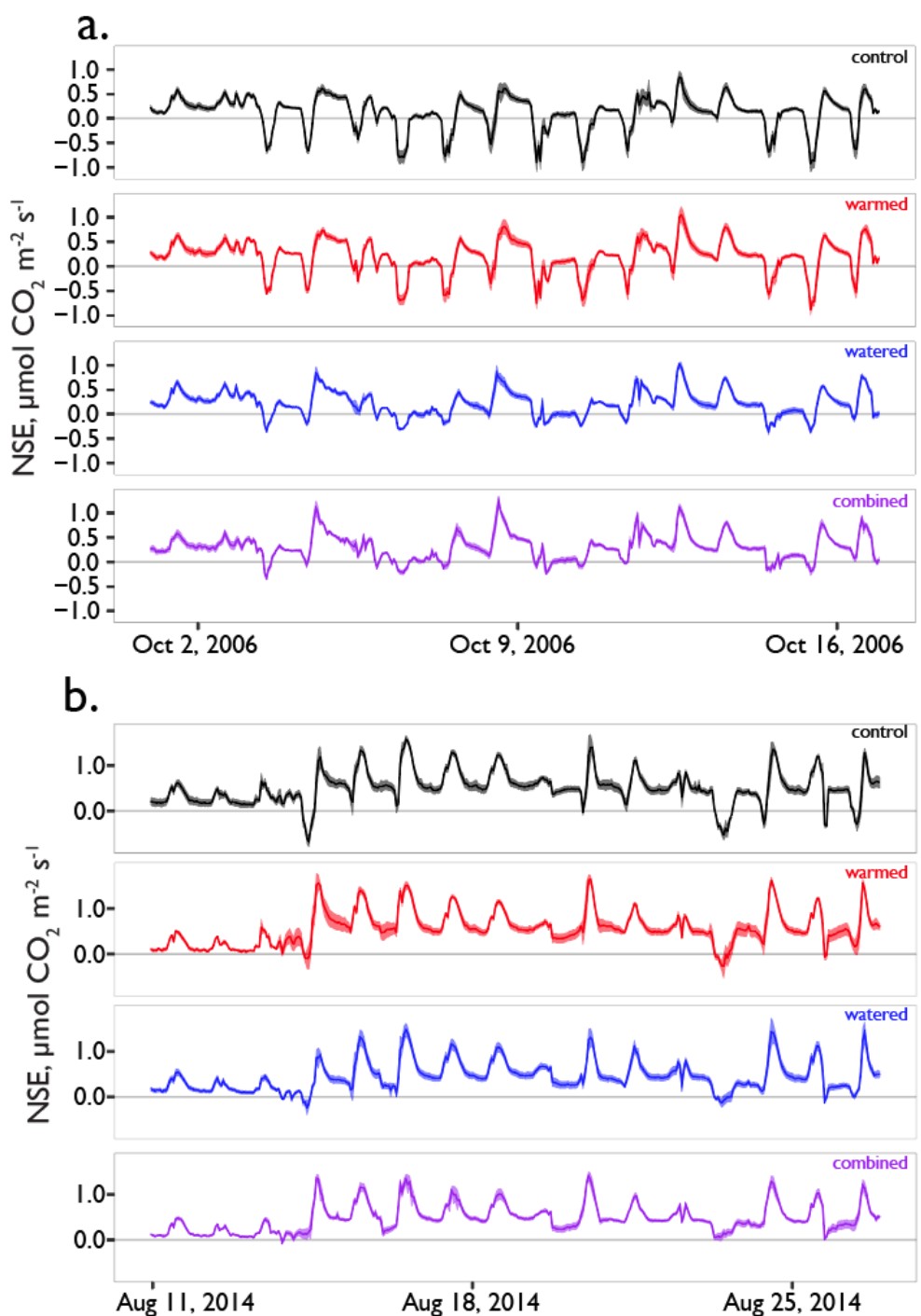


Fig. 5. Examples of hourly $CO_2$ flux patterns during rain events (a) early in the experiment and
(b) in the final season of measurement. Solid lines are the mean and ribbons indicate ± 1 SE. See
Fig. 1a for rainfall patterns at these times.