# Peer review of "Patterns of longer-term climate change effects on CO$_2$ efflux from biocrusted soils differ from those observed in the short-term"

_Biogeosciences, 2017_

## Referee Comment (RC1) · Anonymous Referee #1 · 24 Mar 2018

The study presented by Darrouzet-Nardi et al is an improvement of preliminary results obtained at shorter time span. They analyzed Net Co2 fluxes on soils covered by biocrust under different climate change treatments (warming, watering and a combination of both). They found that these treatments exert a significant effect on CO2 efflux, as consequence of biocrust loss and changes in composition, with important effects on drylands C balance. Overall the manuscript is interesting and well written however there are some points that I would like to discuss before I recomende it publication in Biogeoscience. To correctly contextualize these results, it could be necessary to compare temperature and watering effects with climatic predictions obtained by the different IPCC scenarios. Moreover, I would like to emphasize the potential effect OC

and especially soil labile organic matter on soil respiration. Large effort has been made to explain the potential effect of roots respiration however I recommended talking about soil respiration (including roots, microbial, and other heterotrophs) and the relationship between soil C pools and rain or water pulses (see Lopez-Ballesteros et al., 2016). Watering may promote microbial and decomposition of dead biocrust organisms, with a depletion of labile OC in latter stages (by the cumulative effect of several water pulses, that did not occur in control plots. Moreover, the authors only analyze Net $CO_2$ fluxes but they mention photosynthesis and respiration pulses within the result section (see line 184), I recommended changing it by Net $CO_2$ fixation or release but not respiration and photosynthesis since there is no partition between these two fluxes. Maybe 9 years are not enough for the analysis of a natural (non-induced by the treatment) climatic trend. But I would like to see an exploratory analysis of current climate trend (at least during the study period). This could help to identify any trend in temperature or precipitation that could act in a synergistic manner with experimental treatments Methods section should be improved. There is a reference to a previous study with further details; however there are some key questions that should be explained in the docuemnt: i) I understand that there are a total of 20 plots (5 per treatment). Is it correct? ii) Were biocrust community composition, biomass and coverage of all plots comparable at the beginning of the experiment?. ii) "Water was added in 1.2 mm events manually with backpack sprayers and was applied 40 times from May 31-Sep 20, 2006 and 36 times from June 14-Sep 20 in 2007, with an average time between watering of 2.8 days (Table 1)" According to this sentence, water was added in 2006 and 2007. However as you explain watering was stopped in 2012. Even taken in to account that 2008-2009-2010-2011 were not included in this analysis, this information should be included as it is expected to affect NEE measurements at 2013 and 2014. Did you expect that this could have some effect on respiration patterns observed on 2013-2014. iii) Was the size effect of early warming, watering, and combined treatments on NSE calculated from the random forest models? It is not clear in the clear in the current form. Did you analyze significant differences between treatments?

Some other suggestions Introduction could be better structured by: i) better description of biocrust communities, ii) Firstly describing the importance of drylands in global C fluxes in a context of climate change and then biocrust importance in these ecosystems. Doing this some sentences that are not clear could be better explained (see lines 44-48) Lines 54-58: did you considered the effect of water availability on both process? Positive and negative C fluxes are relative, I would recommend to use C gain and C losses or emissions Fig 2: It could be interesting to show a control plot as figure 2.b. moreover, natural rain pulses could help Fig 3: Are differences significant?

---

## Referee Comment (RC2) · Anonymous Referee #2 · 26 Mar 2018

Dear Editor of Biogeosciences and Dear authors, I have read with interest the manuscript bg-2017-543 entitled:" Patterns of longer-term climate change effects on CO2 efflux from biocrusted soils differ from those observed in the short-term" by Anthony Darrouzet-Nardi et al. The work describe a long term climate change manipulation experiment over soil C fluxes in a semiarid region of the US, the Colorado Plateau, an area with well-known history of active research in the field of biocrusts. I think that the work made and the data provided are novel and interesting, mainly because of the relevance of the methodology used. In this sense, this research line is a step forward in the knowledge of the services that the biocrusts can provide to soils and ecosystems in general. This is because most biocrust gas exchange data available in the literature belong to the isolated crusts or the biocrust isolated with soil attached underneath. Data sets like this give a broader perspective to relevance of biocrusts at the ecosystem level, which is always welcome. Despite of this, there are several points that I would like to comment to authors. The main problems (in general) that I have found with this manuscript are: - An effort should be made in order to clarify the text and data output for potential readers (examples in "detailed comments" section). Something that could be done in this direction is, in the discussion, to add which table or figure of the ones provided in the manuscript are supporting author's findings or explanations. I think that, in order to understand this manuscript properly, readers have to go to too many other related works (Reed et al. 2012; Darrouzet-Nardi et al. 2015; Ferrenberg et al. 2017; Tucker et al 2017…...)

- I think that conclusions linked with the climate manipulation experiment can be analyzed more deeply. This is because I miss: (i) The C fluxes data set between 2008-2012 period (I suppose that the reason behind the lack of them is the huge amount of time necessary to analyze the data, but the gap breaks the possibility of tracking any trend C fluxes-environment, any possibility of analyzing this in the future?) (ii) more detailed information about the macroclimate and the microclimate of the research area between 2006-2014 (Table 1 and supporting information about soil moisture are not enough under my point of view for a work of this dimension) (iii) biocrust coverage information in the control plots and treatments. Is this information provided in other research works? It should be stated somewhere, and (iv) an evaluation of the effect of the pass of the time over the same plots (e.g comparisons in C fluxes and biocrusts covers in controls in 2006 with the same control plot in 2014). I bet that at least part of this information is available at other works of the group and should be easily available for readers. This would improve, under my view, the understanding of the manipulated climate change experiment (which, at the same time, points to interesting results) My detailed comments are: INTRODUCTION - L 45-47: Written that way it seems that soil respiration contribute to C uptake of the soil, please rephrase - L 70-74: Besides the hue included saying that photosynthesis may have a negative relation with temperature, I would suggest to include that the relation between photosynthesis and temperature is normally positively correlated until a saturation trend is found. This contrasts with the pattern in respiration that (as authors say under water availability conditions) is normally positively correlated with T without saturation - L 95: The word "years" was included two times by mistake MATERIAL AND METHODS - L 129: Please quote the source of the prediction - L131: Why did the authors choose the period end of May/beginning of June to mid-September for the wetting of the plots? Are climate change predictions in the area going towards higher precipitation during that period of the year? I think that the reason should be included in the methodology - L145: Which is the frequency of gas exchange measurements inside the chamber during the 3 minutes period that it gets closed? How is the flux exactly calculated at each measuring point? - L152: What about inorganic C fluxes? They are not included in the theoretical balance of NSE. Besides, I think that the NSE concept is interesting and useful for understanding relevance of biocrusts over soil C fluxes, but in order to have a complete understanding of the contribution I think that some data about biocrust coverage should be provided. Which % of the soil surface enclosed by the chamber is covered by BSC at each of the treatments at the beginning of the experiment? And at the end? We know that vascular plants are excluded from the surface, but we do not know any threshold of BSC cover in the plots chosen. - I understand that authors have used statistical methodologies to extrapolate missing data inside the data set together with other methodology to calculate the td comparing controls and treatments. I feel curious about the fact of not having statistical comparison between treatments and controls (e.g, if the effect of the change in the flux in one particular year in the warming treatment (or in any other treatment) is statiscally significant compared with the control) RESULTS - Table 2. I understand that the negative values provided in the table mean that the control mean fluxes where larger than in the treatments, correct? This can be also interpreted as a mean photosynthetic flux, that I think that is not the case. Some clarification about this could be included in legend or text - Figures 1a and 1b are great pieces of information for all the biocrust community, this is a very strong point of this manuscript -

Fig 3, please explain a bit more how were cumulative NSE calculated. Is it possible to include some stats about the differences in NSE created by the treatments? DISCUSSION - L250-270: Authors explain that they found higher C losses in the initial phase of the warming experiment, but that this effect was reversed in the long term. This is interesting, and it seems to me that two possible explanations are given for this (i) reduced soil C availability, which seems to be something like a negative biotic impact due to a legacy of high C loss in the soil and (ii) an effect of lamps drying the soil. I do not know if authors are giving more weight to one or another to explain the results, but after reading with interest the Tucker et al. 2017 paper quoted, a clear question arises: Are the heating lamps affecting the T and moisture of the first milimiters of soil underneath the crusts? (if so, C fluxes would be affected also following Tucker et al.). If the answer is yes, a second one would be, is this creating any experimental bias or the same effect could be expected over the upper mm of soil under a general raise of T between 2-4° as the one assayed? I would welcome a bit of debate about this - L 259-261: The impact of warming over vascular plants photosynthesis is information coming from the quoted papers above, correct? Please modify a bit the sentence to show this clearly - L287-288 What about the evolution of biocrust cover at this experiment in control and experimental plots, let me know please if I am missing any point here. If it is necessary to go to other published paper to see this data (or similar), authors should at least explain it clearly. I would suggest to add a table or figure to summarize this information. - L 294-298: It seems to be important part of the discussion but I do not see the point clearly, please rephrase - L 306-309. Could authors add some type of inputs to show how is the evolution of the NSE in controls under natural environmental conditions during the monitoring (environmental effect over NSE)? Shall we expect a similar or different shift in treatments?

---

## Author Comment (AC1) · 11 Apr 2018

First of all, we thank both reviewers for their comments. They were thoughtful and contained many good suggestions that will improve the manuscript. Reviewer comments are in CAPS and our responses are in normal text.

TO CORRECTLY CONTEXTUALIZE THESE RESULTS, IT COULD BE NECESSARY TO COMPARE TEMPERATURE AND WATERING EFFECTS WITH CLIMATIC PREDICTIONS OBTAINED BY THE DIFFERENT IPCC SCENARIOS.

We will add some discussion of IPCC projections in the Introduction. We agree it could

be helpful. The original proposal for this project was based on this type of information and, as the Reviewer suggests, this would be an appropriate way to contextualize.

MOREOVER, I WOULD LIKE TO EMPHASIZE THE POTENTIAL EFFECT OC AND ESPECIALLY SOIL LABILE ORGANIC MATTER ON SOIL RESPIRATION. LARGE EFFORT HAS BEEN MADE TO EXPLAIN THE POTENTIAL EFFECT OF ROOTS RESPIRATION HOWEVER I RECOMMENDED TALKING ABOUT SOIL RESPIRA-TION (INCLUDING ROOTS, MICROBIAL, AND OTHER HETEROTROPHS) AND THE RELATIONSHIP BETWEEN SOIL C POOLS AND RAIN OR WATER PULSES (SEE LOPEZ-BALLESTEROS ET AL., 2016).

We appreciate this comment and agree that partitioning sources (e.g., heterotrophic consumption of OC vs. root respiration) is important in understanding the mechanisms that drive $CO_2$ exchange with the atmosphere, both now and into the future. With that in mind, the Lopez-Ballesteros et al. 2016 is a very nice exploration of the different components of $CO_2$ efflux. Here we cannot perfectly partition the sources of flux – our goal with this study was to carefully quantify soil $CO_2$ flux with different climates using a manipulative experiment – and thus the mix of eddy covariance, ANPP assessment, and soil chambers as performed by Lopez-Ballesteros et al. 2016 was outside of our experiment's scope. The case we are making here is that, though we are aware of all these sources, the data we have suggest roots are playing a large role in regulating soil flux as observed at the surface. One mistake we made in constructing this argument was referring readers to the appendix in our previous paper (Darrouzet-Nardi et al. Biogeochemistry 2015), where this issue was discussed in detail, including calculations of the size of the organic matter pool and whether it was a plausible source (it is). Instead we should have laid out this logic in this manuscript as well. We didn't want to repeat too much of our previous paper's analysis from there, but we erred too far on the side of redirecting to other papers, which the other review commented on as well. So, we will modify this section to be a more complete and self-contained discussion. Hopefully this will allow a more balanced discussion of sources in this paper to address

this critique.

WATERING MAY PROMOTE MICROBIAL AND DECOMPOSITION OF DEAD BIOCRUST ORGANISMS, WITH A DEPLETION OF LABILE OC IN LATTER STAGES (BY THE CUMULATIVE EFFECT OF SEVERAL WATER PULSES, THAT DID NOT OCCUR IN CONTROL PLOTS. MOREOVER, THE AUTHORS ONLY ANALYZE NET CO2 FLUXES BUT THEY MENTION PHOTOSYNTHESIS AND RESPIRATION PULSES WITHIN THE RESULT SECTION (SEE LINE 184), I RECOMMENDED CHANGING IT BY NET CO2 FIXATION OR RELEASE BUT NOT RESPIRATION AND PHOTOSYNTHESIS SINCE THERE IS NO PARTITION BETWEEN THESE TWO FLUXES.

Assuming this is referring to line 194 at the beginning of the results, we agree, we will change these statements to negative and positive NSE.

MAYBE 9 YEARS ARE NOT ENOUGH FOR THE ANALYSIS OF A NATURAL (NON-INDUCED BY THE TREATMENT) CLIMATIC TREND. BUT I WOULD LIKE TO SEE AN EXPLORATORY ANALYSIS OF CURRENT CLIMATE TREND (AT LEAST DUR-ING THE STUDY PERIOD). THIS COULD HELP TO IDENTIFY ANY TREND IN TEM-PERATURE OR PRECIPITATION THAT COULD ACT IN A SYNERGISTIC MANNER WITH EXPERIMENTAL TREATMENTS.

We agree that more comprehensive descriptions of the climate during the entire study period would be useful to provide and we will both provide these and incorporate them in relevant parts of the discussion.

METHODS SECTION SHOULD BE IMPROVED. THERE IS A REFERENCE TO A PREVIOUS STUDY WITH FURTHER DETAILS; HOWEVER THERE ARE SOME KEY QUESTIONS THAT SHOULD BE EXPLAINED IN THE DOCUEMNT: I) I UNDER-STAND THAT THERE ARE A TOTAL OF 20 PLOTS (5 PER TREATMENT). IS IT CORRECT?

The Reviewer is correct that there are 20 plots, five replicates of each treatment. We do state "The experiment contained five blocks of four treatments each: control, warmed, watered, and combined (warmed + watered) for a total of 20 2 × 2.5 m plots, each of which contained an automated $CO_2$ chamber." We will review the methods to ensure clarity and if there are further suggestions for how we can make the Methods section even more clear (a figure perhaps?), we would be amenable.

WERE BIOCRUST COMMUNITY COMPOSITION, BIOMASS AND COVERAGE OF ALL PLOTS COMPARABLE AT THE BEGINNING OF THE EXPERIMENT?.

They were comparable. We will add a supplemental figure in which we show these data. They have not been reported before so it is a good suggestion for an improvement to the paper.

"WATER WAS ADDED IN 1.2 MM EVENTS MANUALLY WITH BACKPACK SPRAYERS AND WAS APPLIED 40 TIMES FROM MAY 31-SEP 20, 2006 AND 36 TIMES FROM JUNE 14-SEP 20 IN 2007, WITH AN AVERAGE TIME BETWEEN WATERING OF 2.8 DAYS (TABLE 1)" ACCORDING TO THIS SENTENCE, WATER WAS ADDED IN 2006 AND 2007. HOWEVER AS YOU EXPLAIN WATERING WAS STOPPED IN 2012. EVEN TAKEN IN TO ACCOUNT THAT 2008-2009-2010-2011 WERE NOT INCLUDED IN THIS ANALYSIS, THIS INFORMATION SHOULD BE IN-CLUDED AS IT IS EXPECTED TO AFFECT NEE MEASUREMENTS AT 2013 AND 2014. DID YOU EXPECT THAT THIS COULD HAVE SOME EFFECT ON RESPIRA-TION PATTERNS OBSERVED ON 2013-2014.

We will include information on the watering for the years that we left out. This is also a good suggestion for inclusion. It does make sense to report this since it was part of what contributed to what we later call the "legacy" watering treatment and it is not reported elsewhere.

WAS THE SIZE EFFECT OF EARLY WARMING, WATERING, AND COMBINED TREATMENTS ON NSE CALCULATED FROM THE RANDOM FOREST MODELS?

IT IS NOT CLEAR IN THE CLEAR IN THE CURRENT FORM.

The random forest models were only used for gap filling. The effect sizes were calculated using subtraction (treatment - control) and the uncertainty around those differences were calculated using a bootstrap confidence interval. We can include more detail on the bootstrap technique used and we agree we can make it more clear which techniques (random forest vs. bootstrap) were associated with which parts of data processing and analysis.

DID YOU ANALYZE SIGNIFICANT DIFFERENCES BETWEEN TREATMENTS?

Please see our response to the other reviewer on this issue.

INTRODUCTION COULD BE BETTER STRUCTURED BY: I) BETTER DESCRIPTION OF BIOCRUST COMMUNITIES

We agree and can include this information.

II) FIRSTLY DESCRIBING THE IMPORTANCE OF DRYLANDS IN GLOBAL C FLUXES IN A CONTEXT OF CLIMATE CHANGE AND THEN BIOCRUST IMPORTANCE IN THESE ECOSYSTEMS. DOING THIS SOME SENTENCES THAT ARE NOT CLEAR COULD BE BETTER EXPLAINED (SEE LINES 44-48)

The other reviewer noted this section as well and we will work to clarify.

LINES 54-58: DID YOU CONSIDERED THE EFFECT OF WATER AVAILABILITY ON BOTH PROCESS? POSITIVE AND NEGATIVE C FLUXES ARE RELATIVE, I WOULD RECOMMEND TO USE C GAIN AND C LOSSES OR EMISSIONS

In the Wertin et al. 2017 paper we did take into account the effect of water availability (and temperature) on both plant photosynthesis and soil $CO_2$ efflux and the strong relationship we observed between the two fluxes (i.e., photosynthesis and soil $CO_2$ efflux). We postulated that these patterns could have been due to (1) the independent regulation of each flux by climate (i.e., both fluxes were reduced by lowered soil

moisture), (2) reduced soil CO2 efflux being the result of reduced plant C allocation belowground, and thus less root respiration and/or C exudation for heterotrophs, or (3) and/or a mix of both controls.

As far as the terminology, we're not currently understanding the suggestion and would appreciate clarification.

FIG 2: IT COULD BE INTERESTING TO SHOW A CONTROL PLOT AS FIGURE 2.B.

We can add this as suggested.

MOREOVER, NATURAL RAIN PULSES COULD HELP FIG 3: ARE DIFFERENCES SIGNIFICANT?

Annual rainfall is shown in table 1. We could add these to the figure if desired. We are not sure what is meant by "pulses." For the statistical significance issue, please see our comments to reviewer 2.

---

## Author Comment (AC2) · 11 Apr 2018

First of all, we thank both reviewers for their comments. They were thoughtful and contained many good suggestions that will improve the manuscript. Reviewer comments are in CAPS and our responses are in normal text.

AN EFFORT SHOULD BE MADE IN ORDER TO CLARIFY THE TEXT AND DATA OUTPUT FOR POTENTIAL READERS (EXAMPLES IN "DETAILED COMMENTS" SECTION). SOMETHING THAT COULD BE DONE IN THIS DIRECTION IS, IN THE DISCUSSION, TO ADD WHICH TABLE OR FIGURE OF THE ONES PROVIDED IN THE MANUSCRIPT ARE SUPPORTING AUTHOR'S FINDINGS OR EXPLANATIONS.

[Figure]

We can add a few strategic figure/table references in the discussion to help readers. It's a good suggestion.

I THINK THAT, IN ORDER TO UNDERSTAND THIS MANUSCRIPT PROPERLY, READERS HAVE TO GO TO TOO MANY OTHER RELATED WORKS (REED ET AL. 2012; DARROUZET-NARDI ET AL. 2015; FERRENBERG ET AL. 2017; TUCKER ET AL 2017. . ...) -

This is a fair criticism. We will work to make this manuscript more "self-contained."

I THINK THAT CONCLUSIONS LINKED WITH THE CLIMATE MANIPULATION EX-PERIMENT CAN BE ANALYZED MORE DEEPLY. THIS IS BECAUSE I MISS: (I) THE C FLUXES DATA SET BETWEEN 2008-2012 PERIOD (I SUPPOSE THAT THE REA-SON BEHIND THE LACK OF THEM IS THE HUGE AMOUNT OF TIME NECESSARY TO ANALYZE THE DATA, BUT THE GAP BREAKS THE POSSIBILITY OF TRACKING ANY TREND C FLUXES-ENVIRONMENT, ANY POSSIBILITY OF ANALYZING THIS IN THE FUTURE?)

Unfortunately, this is not an analytical gap, it would be manageable to replicate exist-ing statistical analyses for additional years. However, data during these years were not collected due to project logistics and budgets. Keeping the chambers running is a large amount of work and we could not do so for all years of the experiment. It is ex-pensive to run these automated chambers mainly due to the large amount of personnel time they require (weekly physical checks and repairs, frequent data quality monitoring and processing, etc). As it stands, a huge amount of resources were expended to get the data shown here and it would have gone from huge to astronomical to keep them running for all years. As such, the data we have are rare, with other similar studies measuring ∼1-10 times per year as compared to our hourly data (365 * 24 measure-ments per year with some gaps of course). This is why we are able to estimate and contextualize annual totals, which is not possible with other approaches. We believe that this approach provides a valuable point of reference for other studies with spot

measurements. Now with 20/20 hindsight, it would have been ideal to fill in those years with some spot measurements but unfortunately those data were not collected.

MORE DETAILED INFORMATION ABOUT THE MACROCLIMATE AND THE MICRO-CLIMATE OF THE RESEARCH AREA BETWEEN 2006-2014 (TABLE 1 AND SUPPORTING INFORMATION ABOUT SOIL MOISTURE ARE NOT ENOUGH UNDER MY POINT OF VIEW FOR A WORK OF THIS DIMENSION)

We agree and will include much more of this information in our revision.

BIOCRUST COVERAGE INFORMATION IN THE CONTROL PLOTS AND TREATMENTS. IS THIS INFORMATION PROVIDED IN OTHER RESEARCH WORKS? IT SHOULD BE STATED SOMEWHERE,

The other reviewer had the same comment and we agree it should be included. We will include these data.

AN EVALUATION OF THE EFFECT OF THE PASS OF THE TIME OVER THE SAME PLOTS (E.G COMPARISONS IN C FLUXES AND BIOCRUSTS COVERS IN CONTROLS IN 2006 WITH THE SAME CONTROL PLOT IN 2014).

Figures 1 and 3 show the control plot data over time but we can also include statistical comparisons of these plots among years.

L 45-47: WRITTEN THAT WAY IT SEEMS THAT SOIL RESPIRATION CONTRIBUTE TO C UPTAKE OF THE SOIL, PLEASE REPHRASE

Agree this is awkward and we can rephrase.

L 70-74: BESIDES THE HUE INCLUDED SAYING THAT PHOTOSYNTHESIS MAY HAVE A NEGATIVE RELATION WITH TEMPERATURE, I WOULD SUGGEST TO INCLUDE THAT THE RELATION BETWEEN PHOTOSYNTHESIS AND TEMPERATURE IS NORMALLY POSITIVELY CORRELATED UNTIL A SATURATION TREND IS FOUND. THIS CONTRASTS WITH THE PATTERN IN RESPIRATION THAT (AS

AUTHORS SAY UNDER WATER AVAILABILITY CONDITIONS) IS NORMALLY POSI-TIVELY CORRELATED WITH T WITHOUT SATURATION

Our data suggest that both photosynthesis and respiration reach a threshold with temperature in which the positive relationship is reversed. It may not occur at the same temperature. We agree that juxtaposing this against wetter systems where temperature effects are often positive is a great idea and we can work to make this more clear.

L 129: PLEASE QUOTE THE SOURCE OF THE PREDICTION

Good point. We will include.

L131: WHY DID THE AUTHORS CHOOSE THE PERIOD END OF MAY/BEGINNING OF JUNE TO MID-SEPTEMBER FOR THE WETTING OF THE PLOTS? ARE CLIMATE CHANGE PREDICTIONS IN THE AREA GOING TOWARDS HIGHER PRECIPITATION DURING THAT PERIOD OF THE YEAR? I THINK THAT THE REASON SHOULD BE INCLUDED IN THE METHODOLOGY

The reason is that this is mostly a spring rain system, with variable monsoons later in the year that are often insubstantial. As such, the more reliable spring rain makes spring the main growing season for plants. We will clarify this.

L145: WHICH IS THE FREQUENCY OF GAS EXCHANGE MEASUREMENTS INSIDE THE CHAMBER DURING THE 3 MINUTES PERIOD THAT IT GETS CLOSED? HOW IS THE FLUX EXACTLY CALCULATED AT EACH MEASURING POINT?

We will include these technical details.

L152: WHAT ABOUT INORGANIC C FLUXES? THEY ARE NOT INCLUDED IN THE THEORETICAL BALANCE OF NSE. BESIDES, I THINK THAT THE NSE CONCEPT IS INTERESTING AND USEFUL FOR UNDERSTANDING RELEVANCE OF BIOCRUSTS OVER SOIL C FLUXES, BUT IN ORDER TO HAVE A COMPLETE UNDERSTANDING OF THE CONTRIBUTION I THINK THAT SOME DATA ABOUT BIOCRUST COVERAGE SHOULD BE PROVIDED. WHICH % OF THE SOIL SUR-

FACE ENCLOSED BY THE CHAMBER IS COVERED BY BSC AT EACH OF THE TREATMENTS AT THE BEGINNING OF THE EXPERIMENT? AND AT THE END? WE KNOW THAT VASCULAR PLANTS ARE EXCLUDED FROM THE SURFACE, BUT WE DO NOT KNOW ANY THRESHOLD OF BSC COVER IN THE PLOTS CHOSEN.

As stated above, we will include information on biocrust coverage. Inorganic C fluxes could be playing a role but in our previous study we describe why we think it is probably not a large factor. A major goal for the revision will be to make this information more self-contained in this manuscript so we will touch on this as well.

I UNDERSTAND THAT AUTHORS HAVE USED STATISTICAL METHODOLOGIES TO EXTRAPOLATE MISSING DATA INSIDE THE DATA SET TOGETHER WITH OTHER METHODOLOGY TO CALCULATE THE TD COMPARING CONTROLS AND TREATMENTS. I FEEL CURIOUS ABOUT THE FACT OF NOT HAVING STATISTICAL COMPARISON BETWEEN TREATMENTS AND CONTROLS (E.G, IF THE EFFECT OF THE CHANGE IN THE FLUX IN ONE PARTICULAR YEAR IN THE WARMING TREATMENT (OR IN ANY OTHER TREATMENT) IS STATISCALLY SIGNIFICANT COMPARED WITH THE CONTROL)

There is a statistical comparison but we focus on effect sizes instead of statistical "significance." The cited Nakagawa and Cuthill paper provides a good justification for this approach. We calculate uncertainty surrounding td using bootstrapped confidence intervals. Though we have explicitly avoided using the null hypothesis statistical testing (NHST) paradigm, we note that as a heuristic, confidence intervals that do not contain 0 would be marked as "significant" in NHST. Thus, confidence intervals provide more complete information as compared to what a p-value would provide: they constrain effect size with bounds instead of only telling us the probability that that bounds on the effect size contain zero. Tukey (1991) "Philosophy of multiple comparisons" provides another strong and concise argument on why significance testing is too black and white. As an example, both the -11.8 [-21.7, 0.4] and the -1.2 [20.3, -15.1] would technically be "not significant" but these are different results that warrant different interpretations.

The first implies a much greater likelihood that the effect is in the direction of less C flux in the control and could well be of substantial quantity whereas the second implies poor constraint and lack of good information on the effect size due to high variability among chambers. It could be high, low, or negligible, with greater sample size needed for better constraints.

TABLE 2. I UNDERSTAND THAT THE NEGATIVE VALUES PROVIDED IN THE TABLE MEAN THAT THE CONTROL MEAN FLUXES WHERE LARGER THAN IN THE TREATMENTS, CORRECT? THIS CAN BE ALSO INTERPRETED AS A MEAN PHOTOSYNTHETIC FLUX, THAT I THINK THAT IS NOT THE CASE. SOME CLARIFICATION ABOUT THIS COULD BE INCLUDED IN LEGEND OR TEXT.

Yes, it is correct that this is the difference between control and treatment plots and is not indicative of photosynthesis. We can clarify in the caption, which we will do. All plots showed net C losses to the atmosphere at all times - but some of them more so than others.

FIG 3, PLEASE EXPLAIN A BIT MORE HOW WERE CUMULATIVE NSE CALCULATED. IS IT POSSIBLE TO INCLUDE SOME STATS ABOUT THE DIFFERENCES IN NSE CREATED BY THE TREATMENTS?

The stats from table 2 are directly associated with these numbers. One option would be to graphically display the confidence intervals from table 2 in a separate panel. If the Editors or Reviewers would prefer that please, let us know. If the Reviewer is suggesting NHST-based graphical components such as significance stars or letters, we would prefer not to for the reasons stated above.

L250-270: AUTHORS EXPLAIN THAT THEY FOUND HIGHER C LOSSES IN THE INITIAL PHASE OF THE WARMING EXPERIMENT, BUT THAT THIS EFFECT WAS REVERSED IN THE LONG TERM. THIS IS INTERESTING, AND IT SEEMS TO ME THAT TWO POSSIBLE EXPLANATIONS ARE GIVEN FOR THIS (I) REDUCED SOIL C AVAILABILITY, WHICH SEEMS TO BE SOMETHING LIKE A NEGATIVE BIOTIC

IMPACT DUE TO A LEGACY OF HIGH C LOSS IN THE SOIL AND (II) AN EFFECT OF LAMPS DRYING THE SOIL. I DO NOT KNOW IF AUTHORS ARE GIVING MORE WEIGHT TO ONE OR ANOTHER TO EXPLAIN THE RESULTS, BUT AFTER READING WITH INTEREST THE TUCKER ET AL. 2017 PAPER QUOTED, A CLEAR QUESTION ARISES: ARE THE HEATING LAMPS AFFECTING THE T AND MOISTURE OF THE FIRST MILIMITERS OF SOIL UNDERNEATH THE CRUSTS? (IF SO, C FLUXES WOULD BE AFFECTED ALSO FOLLOWING TUCKER ET AL.). IF THE ANSWER IS YES, A SECOND ONE WOULD BE, IS THIS CREATING ANY EXPERIMENTAL BIAS OR THE SAME EFFECT COULD BE EXPECTED OVER THE UPPER MM OF SOIL UNDER A GENERAL RAISE OF T BETWEEN 2-4âŮę AS THE ONE ASSAYED? I WOULD WELCOME A BIT OF DEBATE ABOUT THIS

We agree that this is one of the crucial details in interpreting these data and we did try to address the point, but in reading back over what we submitted, more could be done to illuminate this issue. We will work to do this in the revision. Our basic argument was that we in fact do not see obvious moisture differences among treatments (the supplemental figure provides some information on this). However, the Tucker paper implies that our current measurement may have been missing the surface moisture dynamics. In addition, it is to be expected that the change in community composition of the crusts will play some role. So we think both likely play a role and unfortunately from this study, we can't totally disentangle which is more important. The inclusion of the biocrust cover data, which we should have provided will help to inform this for the reader as well.

L 259- 261: THE IMPACT OF WARMING OVER VASCULAR PLANTS PHOTOSYNTHESIS IS INFORMATION COMING FROM THE QUOTED PAPERS ABOVE, CORRECT? PLEASE MODIFY A BIT THE SENTENCE TO SHOW THIS CLEARLY

We can certainly do this. Thanks for pointing out the lack of clarity here.

L287-288 WHAT ABOUT THE EVOLUTION OF BIOCRUST COVER AT THIS EXPER-

IMENT IN CONTROL AND EXPERIMENTAL PLOTS, LET ME KNOW PLEASE IF I AM MISSING ANY POINT HERE. IF IT IS NECESSARY TO GO TO OTHER PUBLISHED PAPER TO SEE THIS DATA (OR SIMILAR), AUTHORS SHOULD AT LEAST EXPLAIN IT CLEARLY. I WOULD SUGGEST TO ADD A TABLE OR FIGURE TO SUMMARIZE THIS INFORMATION.

Both Reviewers said this and we agree with them. This will be in the next version.

L 294-298: IT SEEMS TO BE IMPORTANT PART OF THE DISCUSSION BUT I DO NOT SEE THE POINT CLEARLY, PLEASE REPHRASE.

We agree this is not as clear as it could be and we will work to rephrase.

L 306-309. COULD AUTHORS ADD SOME TYPE OF INPUTS TO SHOW HOW IS THE EVOLUTION OF THE NSE IN CONTROLS UNDER NATURAL ENVIRONMEN-TAL CONDITIONS DURING THE MONITORING (ENVIRONMENTAL EFFECT OVER NSE)? SHALL WE EXPECT A SIMILAR OR DIFFERENT SHIFT IN TREATMENTS?

Environmental associations with NSE is something we addressed in our previous paper and we agree that summarizing the expected effect of moisture and temperature based on these relationships would be helpful here. We can add some expectations of temperature and moisture effects based on this.

―――――――――――――――――

---

## Author Response (AR1)

We thank both reviewers for their comments. They were thoughtful and contained many good suggestions that have improved the manuscript. Below we sorted comments from both reviewers into major categories and describe the changes we made based on the comments.

**SELF CONTAINMENT**

*METHODS SECTION SHOULD BE IMPROVED. THERE IS A REFERENCE TO A PREVIOUS STUDY WITH FURTHER DETAILS; HOWEVER THERE ARE SOME KEY QUESTIONS THAT SHOULD BE EXPLAINED IN THE DOCUEMNT: I) I UNDERSTAND THAT THERE ARE A TOTAL OF 20 PLOTS (5 PER TREATMENT). IS IT CORRECT?*

The Reviewer is correct that there are 20 plots, five replicates of each treatment. We reworded this section of the methods to make this more clear and tried to describe all relevant information in all cases where we referenced other studies.

*I THINK THAT, IN ORDER TO UNDERSTAND THIS MANUSCRIPT PROPERLY, READERS HAVE TO GO TO TOO MANY OTHER RELATED WORKS (REED ET AL. 2012; DARROUZET-NARDI ET AL. 2015; FERRENBERG ET AL. 2017; TUCKER ET AL 2017. . . ...) -* This is a fair criticism. We worked to make this manuscript more "self-contained" in numerous places throughout the manuscript

*MOREOVER, I WOULD LIKE TO EMPHASIZE THE POTENTIAL EFFECT OC AND ESPECIALLY SOIL LABILE ORGANIC MATTER ON SOIL RESPIRATION. LARGE EFFORT HAS BEEN MADE TO EXPLAIN THE POTENTIAL EFFECT OF ROOTS RESPIRATION HOWEVER I RECOMMENDED TALKING ABOUT SOIL RESPIRATION (INCLUDING ROOTS, MICROBIAL, AND OTHER HETEROTROPHS) AND THE RELATIONSHIP BETWEEN SOIL C POOLS AND RAIN OR WATER PULSES (SEE LOPEZ-BALLESTEROS ET AL., 2016).*

We appreciate this comment and agree that partitioning sources (e.g., heterotrophic consumption of OC vs. root respiration) is important in understanding the mechanisms that drive $CO_2$ exchange with the atmosphere, both now and into the future. With that in mind, the Lopez-Ballesteros et al. 2016 is a very nice exploration of the different components of $CO_2$ efflux. Here we cannot perfectly partition the sources of flux – our goal with this study was to carefully quantify soil $CO_2$ flux with different climates using a manipulative experiment – and thus the mix of eddy covariance, ANPP assessment, and soil chambers as performed by Lopez-Ballesteros et al. 2016 was outside of our experiment's scope. The case we are making here is that, though we are aware of all these sources, the data we have suggest roots are playing a large role in regulating soil flux as observed at the surface. One mistake we made in constructing this argument was referring readers to the appendix in our previous paper (Darrouzet-Nardi et al. Biogeochemistry 2015), where this issue was discussed in detail, including calculations of the size of the organic matter pool and whether it was a plausible source (it is). Instead we have now laid out this logic in this manuscript as well. We didn't want to repeat too much of our previous paper's analysis from there, but we erred too far on the side of redirecting to other papers, which the other review commented on as well. We have largely rewritten this section (discussion section 4.3) to be a more complete and self-contained discussion. Hopefully this will allow a more balanced discussion of sources in this paper to address this critique.

*L152: WHAT ABOUT INORGANIC C FLUXES? THEY ARE NOT INCLUDED IN THE*
*THEORETICAL BALANCE OF NSE.*
We did some further reading on this and decided we in fact cannot rule inorganic C fluxes out as
playing a role so we have included a paragraph on the issue.
*L250-270: AUTHORS EXPLAIN THAT THEY FOUND HIGHER C LOSSES IN THE INITIAL*
*PHASE OF THE WARMING EXPERIMENT, BUT THAT THIS EFFECT WAS REVERSED IN*
*THE LONG TERM. THIS IS INTERESTING, AND IT SEEMS TO ME THAT TWO POSSIBLE*
*EXPLANATIONS ARE GIVEN FOR THIS (I) REDUCED SOIL C AVAILABILITY, WHICH*
*SEEMS TO BE SOMETHING LIKE A NEGATIVE BIOTIC IMPACT DUE TO A LEGACY OF*
*HIGH C LOSS IN THE SOIL AND (II) AN EFFECT OF LAMPS DRYING THE SOIL. I DO NOT*
*KNOW IF AUTHORS ARE GIVING MORE WEIGHT TO ONE OR ANOTHER TO EXPLAIN*
*THE RESULTS, BUT AFTER READING WITH INTEREST THE TUCKER ET AL. 2017 PAPER*
*QUOTED, A CLEAR QUESTION ARISES: ARE THE HEATING LAMPS AFFECTING THE T*
*AND MOISTURE OF THE FIRST MILIMITERS OF SOIL UNDERNEATH THE CRUSTS? (IF*
*SO, C FLUXES WOULD BE AFFECTED ALSO FOLLOWING TUCKER ET AL.). IF THE*
*ANSWER IS YES, A SECOND ONE WOULD BE, IS THIS CREATING ANY EXPERIMENTAL*
*BIAS OR THE SAME EFFECT COULD BE EXPECTED OVER THE UPPER MM OF SOIL*
*UNDER A GENERAL RAISE OF T BETWEEN 2-4∘ AS THE ONE ASSAYED? I WOULD*
*WELCOME A BIT OF DEBATE ABOUT THIS*
We agree that this is one of the crucial details in interpreting these data and we did try to address
the point more completely in the revision. Our basic argument was that we in fact do not see
obvious moisture differences among treatments (the supplemental figure provides some
information on this). However, the Tucker paper implies that our current measurement may have
been missing the surface moisture dynamics. In addition, it is to be expected that the change in
community composition of the crusts will play some role. So we think both likely play a role and
unfortunately from this study, we can't totally disentangle which is more important. The
inclusion of the biocrust cover data helps to inform on this matter as well. As for any anomalous
heating effects on the surface of the soil creating a bias, we are not aware of such an effect and
believe that the heating simulates within reason what future conditions may be like in these
ecosystems.
**ADDED DATA AND ANALYSIS**
**Add biocrust composition data**
*-WERE BIOCRUST COMMUNITY COMPOSITION, BIOMASS AND COVERAGE OF ALL*
*PLOTS COMPARABLE AT THE BEGINNING OF THE EXPERIMENT?.*
*-BIOCRUST COVERAGE INFORMATION IN THE CONTROL PLOTS AND TREATMENTS. IS*
*THIS INFORMATION PROVIDED IN OTHER RESEARCH WORKS? IT SHOULD BE STATED*
*SOMEWHERE,*

*-L287-288 WHAT ABOUT THE EVOLUTION OF BIOCRUST COVER AT THIS EXPERIMENT*
*IN CONTROL AND EXPERIMENTAL PLOTS, LET ME KNOW PLEASE IF I AM MISSING*
*ANY POINT HERE. IF IT IS NECESSARY TO GO TO OTHER PUBLISHED PAPER TO SEE*
*THIS DATA (OR SIMILAR), AUTHORS SHOULD AT LEAST EXPLAIN IT CLEARLY. I*
*WOULD SUGGEST TO ADD A TABLE OR FIGURE TO SUMMARIZE THIS INFORMATION.*
*-BESIDES, I THINK THAT THE NSE CONCEPT IS INTERESTING AND USEFUL FOR*
*UNDERSTANDING RELEVANCE OF BIOCRUSTS OVER SOIL C FLUXES, BUT IN ORDER*
*TO HAVE A COMPLETE UNDERSTANDING OF THE CONTRIBUTION I THINK THAT*
*SOME DATA ABOUT BIOCRUST COVERAGE SHOULD BE PROVIDED. WHICH % OF THE*
*SOIL SURFACE ENCLOSED BY THE CHAMBER IS COVERED BY BSC AT EACH OF THE*
*TREATMENTS AT THE BEGINNING OF THE EXPERIMENT? AND AT THE END? WE*
*KNOW THAT VASCULAR PLANTS ARE EXCLUDED FROM THE SURFACE, BUT WE DO*
*NOT KNOW ANY THRESHOLD OF BSC COVER IN THE PLOTS CHOSEN.*
We added a figure in which we show these data and discussed them throughout the manuscript.
This has improved the manuscript and we thank the reviewers for the suggestion.
*INTRODUCTION COULD BE BETTER STRUCTURED BY: I) BETTER DESCRIPTION OF*
*BIOCRUST COMMUNITIES*
We added some biocrust species cover info in the introduction.
**Add a control plot to figure 2**
*FIG 2: IT COULD BE INTERESTING TO SHOW A CONTROL PLOT AS FIGURE 2.B.*
We added this as suggested.
**Assemble climate data for whole study period**
*-MAYBE 9 YEARS ARE NOT ENOUGH FOR THE ANALYSIS OF A NATURAL (NON-*
*INDUCED BY THE TREATMENT) CLIMATIC TREND. BUT I WOULD LIKE TO SEE AN*
*EXPLORATORY ANALYSIS OF CURRENT CLIMATE TREND (AT LEAST DURING THE*
*STUDY PERIOD). THIS COULD HELP TO IDENTIFY ANY TREND IN TEMPERATURE OR*
*PRECIPITATION THAT COULD ACT IN A SYNERGISTIC MANNER WITH EXPERIMENTAL*
*TREATMENTS.*
*-MORE DETAILED INFORMATION ABOUT THE MACROCLIMATE AND THE*
*MICROCLIMATE OF THE RESEARCH AREA BETWEEN 2006-2014 (TABLE 1 AND*
*SUPPORTING INFORMATION ABOUT SOIL MOISTURE ARE NOT ENOUGH UNDER MY*
*POINT OF VIEW FOR A WORK OF THIS DIMENSION)*
We added substantially more climate information to the figures and results.
*"WATER WAS ADDED IN 1.2 MM EVENTS MANUALLY WITH BACKPACK SPRAYERS AND*
*WAS APPLIED 40 TIMES FROM MAY 31-SEP 20, 2006 AND 36 TIMES FROM JUNE 14-SEP*
*20 IN 2007, WITH AN AVERAGE TIME BETWEEN WATERING OF 2.8 DAYS (TABLE 1)"*

*ACCORDING TO THIS SENTENCE, WATER WAS ADDED IN 2006 AND 2007. HOWEVER AS*
*YOU EXPLAIN WATERING WAS STOPPED IN 2012. EVEN TAKEN IN TO ACCOUNT THAT*
*2008-2009-2010-2011 WERE NOT INCLUDED IN THIS ANALYSIS, THIS INFORMATION*
*SHOULD BE INCLUDED AS IT IS EXPECTED TO AFFECT NEE MEASUREMENTS AT 2013*
*AND 2014. DID YOU EXPECT THAT THIS COULD HAVE SOME EFFECT ON*
*RESPIRATION PATTERNS OBSERVED ON 2013-2014.*

We included information on the watering for the years that we left out.

**Add extra statistical comparisons among time periods (Compare expected change in**
**temp/moisture based on correlations through time with what we saw)**
*-AN EVALUATION OF THE EFFECT OF THE PASS OF THE TIME OVER THE SAME PLOTS*
*(E.G COMPARISONS IN C FLUXES AND BIOCRUSTS COVERS IN CONTROLS IN 2006*
*WITH THE SAME CONTROL PLOT IN 2014).*

*-L 306-309. ADD INFORMATION ON CHANGES IN NSE IN CONTROLS UNDER NATURAL*
*ENVIRONMENTAL CONDITIONS OVER THE COURSE OF THE STUDY. SHALL WE*
*EXPECT A SIMILAR OR DIFFERENT SHIFT IN TREATMENTS?*

We added a supplementary table showing effect sizes for change within treatments over time and
discussed it in the results.

**DESCRIPTION OF STATISTICS**

**Improve description of statistics**
*WAS THE SIZE EFFECT OF EARLY WARMING, WATERING, AND COMBINED*
*TREATMENTS ON NSE CALCULATED FROM THE RANDOM FOREST MODELS? IT IS NOT*
*CLEAR IN THE CLEAR IN THE CURRENT FORM.*

The random forest models were only used for gap filling. The effect sizes were calculated using
subtraction (treatment - control) and the uncertainty around those differences were calculated
using a confidence intervals. We did a couple of things here. First, we include switched to a more
straightforward technique used to calculate the confidence intervals (mixed effects models).
Changes to the results were negligible but the analyses are more reproducible and standard. We
also worked to make it more clear which techniques (random forest vs. mixed effects models)
were associated with which parts of data processing and analysis.

*I UNDERSTAND THAT AUTHORS HAVE USED STATISTICAL METHODOLOGIES TO*
*EXTRAPOLATE MISSING DATA INSIDE THE DATA SET TOGETHER WITH OTHER*
*METHODOLOGY TO CALCULATE THE TD COMPARING CONTROLS AND TREATMENTS.*
*I FEEL CURIOUS ABOUT THE FACT OF NOT HAVING STATISTICAL COMPARISON*
*BETWEEN TREATMENTS AND CONTROLS (E.G, IF THE EFFECT OF THE CHANGE IN*
*THE FLUX IN ONE PARTICULAR YEAR IN THE WARMING TREATMENT (OR IN ANY*
*OTHER TREATMENT) IS STATISCALLY SIGNIFICANT COMPARED WITH THE CONTROL)*

*-FIG 3, PLEASE EXPLAIN A BIT MORE HOW WERE CUMULATIVE NSE CALCULATED. IS*
*IT POSSIBLE TO INCLUDE SOME STATS ABOUT THE DIFFERENCES IN NSE CREATED*
*BY THE TREATMENTS?*
There is a statistical comparison but we focus on effect sizes instead of statistical "significance."
The cited Nakagawa and Cuthill paper provides a good justification for this approach. We
calculate uncertainty surrounding *td* using confidence intervals. Though we have explicitly
avoided using the null hypothesis statistical testing (NHST) paradigm, we note that as a heuristic,
confidence intervals that do not contain 0 would be marked as "significant" in NHST. Thus,
confidence intervals provide more complete information as compared to what a p-value would
provide: they constrain effect size with bounds instead of only telling us the probability that that
bounds on the effect size contain zero. Tukey (1991) "Philosophy of multiple comparisons"
provides another strong and concise argument on why significance testing is too black and white.
As an example, both the -11.8 [-21.7, 0.4] and the -1.2 [20.3, -15.1] would technically be "not
significant" but these are different results that warrant different interpretations. The first implies
a much greater likelihood that the effect is in the direction of less C flux in the control and could
well be of substantial quantity whereas the second implies poor constraint and lack of good
information on the effect size due to high variability among chambers. It could be high, low, or
negligible, with greater sample size needed for better constraints.
**Make sure reason for watering is included**
*L131: WHY DID THE AUTHORS CHOOSE THE PERIOD END OF MAY/BEGINNING OF*
*JUNE TO MID-SEPTEMBER FOR THE WETTING OF THE PLOTS? ARE CLIMATE*
*CHANGE PREDICTIONS IN THE AREA GOING TOWARDS HIGHER PRECIPITATION*
*DURING THAT PERIOD OF THE YEAR? I THINK THAT THE REASON SHOULD BE*
*INCLUDED IN THE METHODOLOGY*
The reason is that this was based on predictions of greater precipitation frequency during the
monsoon season. We have added this reasoning into the manuscript.
**Technical details on chamber measurement**
*L145: WHICH IS THE FREQUENCY OF GAS EXCHANGE MEASUREMENTS INSIDE THE*
*CHAMBER DURING THE 3 MINUTES PERIOD THAT IT GETS CLOSED? HOW IS THE*
*FLUX EXACTLY CALCULATED AT EACH MEASURING POINT?*
We added these technical details.

[revised manuscript text omitted]

---

## Author Response (AR2)

We thank the reviewers again for their further comments and have addressed each of them below.

Reviewer #1:

Lines 383-382: Thought artificial wetting information is included within the table, I still miss some information in the text. One have to check table to understand that wetting continues after 2007. One sentence could be nice

*We agree with the reviewer and have added this (L219)*

In figure 3 I still have the impression that there is a decline in Organic carbon available for decomposition (Maybe death biocrust material) that progressively limited CO2 puffs after wetting treatments. What do you think about this issue?

*We state that "the decline in the size of the "puffs" of CO2 that were associated with the 1.2 mm watering treatments are likely driven by declines in biocrust activity (Fig. 3), as these small watering events primarily affect the surface of the soil." We added the following sentence: "These biocrust activities could include both biocrust respiration and decomposition of dead biocrust material." We agree this could be part of what is happening.*

Reviewer #2:

Authors should clarify more in the stats section of the methodology when comparisons are significant and when not (I mean, the fact of the interval including the 0 having a statistical meaning). Some readers may not be used to the statistical approach made. Besides, please explain what is exactly the block as random effect in the mixed effects linear model.

*We pointed out the connection between p-values and confidence intervals to show that confidence intervals provide a superset of information in comparison to p-values. They are also more easily interpreted. It's the difference between 'treatment A is 10±5% bigger than treatment B' vs. 'treatment A is bigger than treatment B (P<0.05).' The former is always more informative in that it implies the latter while also giving a quantitative estimate of the effect size. We do not want readers to back-convert our results to an NHST framework.*

*We agree with the reviewer that the description of blocking could be improved and we have done so. We added "grouped into 5 blocks determined by spatial location on the hillslope" in the study site section (L199).*

I think that table S2 could be improved, information provided there is quite interesting and I see it unclear. For example, the table is named as "differences within treatments over time", but I do not see clear what is being compared and which the time lapse of the comparison is. Please include this information in the figure legend, which should make the figure independent.

*Yes, those treatments should have been spelled out instead of listed as codes and the time period was left out. We have corrected this.*

Please provide sources for new data included in the discussion (Ls 672-676)
*This is unpublished data from our own group. We changed this to "S.C. Reed, unpublished data"*
*If Biogeosciences would prefer another format, please let us know.*
New figure 1 is quite informative and a good improvement, but one question arises which I think
that can be of interest and could be worthy to be considered in the discussion. Looking at the
results of the controls, it seems that there is a natural pattern of involution in the biocrust
succession (going towards a higher cyano cover) between 2005-2017. This is especially striking
in lichens, with treatments and controls showing no differences between them. Any idea or input
about these patterns and the possible links with C fluxes?
*It's a good point and we agree we should elaborate. We have substantially reworked one of the*
*discussion paragraphs to take this better into account. See lines 367-379.*

[revised manuscript text omitted]